# Cosmic-Ray neutron Sensor PYthon tool (crspy 1.2.1): An open-source tool for the processing of cosmic-ray neutron and soil moisture data

Daniel Power[1], Miguel Angel Rico-Ramirez[1], Sharon Desilets[2], Darin Desilets[2], Rafael Rosolem[1,3]

[1] Faculty of Engineering, University of Bristol, Bristol, UK
[2] Hydroinnova, Albuquerque, New Mexico, USA
[3] Cabot Institute for the Environment, University of Bristol, Bristol, UK

*Correspondence to*: Daniel Power (daniel.power@bristol.ac.uk)

**Abstract.**

Understanding soil moisture dynamics at the sub-kilometre scale is increasingly important especially with continuous development of hyper-resolution land-surface and hydrological models. Cosmic Ray Neutron Sensors (CRNS) are able to provide estimates of soil moisture at this elusive scale and networks of these sensors have been expanding across the world over the previous decade. However, each network currently implements its own protocol when processing raw data into soil moisture estimates. As a consequence, this lack of a harmonized global dataset can ultimately lead to limitations in the global assessment of the CRNS technology from multiple networks. Here we present crspy, an open-source python tool that is designed to facilitate the processing of raw CRNS data into soil moisture estimates in an easy and harmonized way. We outline the basic structure of this tool discussing the correction methods used as well as discussing the metadata that crspy can create about each site. Metadata can add value to global scale studies of field scale soil moisture estimates by providing additional routes to understanding catchment similarities and differences. We demonstrate that current differences in processing methodologies can lead to misinterpretations when comparing sites from different networks and having a tool to provide a harmonized dataset can help to mitigate these issues. By being open source, crspy can also serve as a development and testing tool for new understanding of the CRNS technology as well as being used as a teaching tool for the community.

## 1 Context and Background

Soil moisture exerts a large influence on hydrological (Van Loon et al., 2015), biogeochemical (Schlesinger et al., 2015), and climatic processes (Dobriyal et al., 2012; Koster et al., 2004), agricultural systems (Fontanet et al., 2018; Dutta et al., 2014), landslide modelling (Zhuo et al., 2019) and earth system sciences (Fang and Lakshmi, 2014; Bonan, 2008). Its accurate measurement is important to advance our understanding of these areas of research. In-situ point scale soil moisture estimates, such as Time Domain Reflectometry (TDR), can provide higher temporal resolution; however spatial resolution is still limited, on the order of centimetres. Soil heterogeneity can lead to uncertainties when upscaling to the field scale (Western et al., 1999), which would be required for regional or larger scale hydrological modelling. Alternatively, satellite remote sensing products such as Soil Moisture Active Passive (SMAP) and Soil Moisture and Ocean Salinity (SMOS) can provide global estimates of

soil moisture at a coarser spatial (~40km resolution) and temporal (~3 days) scale, and at much shallower depths (~5cm) (Entekhabi et al., 2010; Kerr et al., 2001). It is accepted that we will require a finer spatial resolution than currently achievable through remote sensing estimates for tasks such as increasing our understanding of sub-kilometre land-atmosphere interactions,

or towards the improvements of farming practices (such as through the process of irrigation scheduling), and so there is a need for additional processing of ancillary data for the downscaling of these products (Portal et al., 2020; Alemohammad et al., 2018). In addition, the recent push for hyper resolution global modelling means we require measurements at a finer spatial resolution, on the order of sub-kilometer scales (Wood et al., 2011). Bierkens et al., (2015) discussed the implications of moving from a more standard resolution ~50km model to a hyper-resolution model that is sub-kilometre. The study further

discussed the need to move from sub-grid paradigms, that represent a conceptualized form of earth system dynamics from within the standard 50km resolution model, to explicit dynamics of earth system processes at scales < 50km. This requires a greater understanding of environmental functions at sub-kilometre, spatial scales, which in turn requires accurate measurements of environment states at the same scales.

Cosmic-Ray Neutron Sensors (CRNS) are a relatively new technology that allows estimates of soil moisture at the field scale (~600m diameter) at hourly temporal resolution. Zreda et al., (2008) demonstrated that fast neutrons are mainly moderated by hydrogen atoms, which allows us to infer changes in water content in the soil profile. A tube attached to the sensor, filled with a gas such as helium or boron trifluoride, is able to detect fast neutrons that pass through it by inducing a voltage difference. Desilets et al., (2010) introduced an equation used to convert neutron counting rates into gravimetric soil moisture which has

been further improved upon by Dong et al., (2014) and Hawdon et al., (2014) (equation 1). The original equation along with the further advancements provides us with estimates of volumetric soil moisture:

$$\theta_{vol} = \left[ \frac{a_0}{\frac{N_{raw} \cdot f_p \cdot f_i \cdot f_h \cdot f_v}{N_0} - a_1} - a_2 - LW - WSOM \right] \frac{\rho_{bd}}{\rho w} \tag{1}$$

where $\theta_{vol}$ is volumetric soil moisture (cm$^3$/cm$^3$); $a_0, a_1$, and $a_2$ are coefficients obtained from neutron particle physics modelling (Zreda et al., 2008; Desilets et al., 2010) and assumed to be constants; $LW$ is the lattice (chemically-bounded mineral) water (g g $^{-1}$), $WSOM$ is the water equivalent of soil organic carbon (g of water per g of soil), $\rho_{bd}$ is the bulk density of the dry soil (g/cm$^3$) and $\rho w$ is the density of water defined as 1 g/cm$^3$. $N_{raw}$ is the measured raw, uncorrected, neutron count identified over the given integration time, usually set to one hour. $f_p$, $f_i$, $f_h$, and $f_v$ represent correction factors applied to

$N_{raw}$ to account for additional influences on the neutron signal other than soil moisture; they are corrections for air pressure, incoming neutron intensity, atmospheric water vapour and above ground biomass, respectively. $N_0$ is the theoretical neutron count found in absolutely dry conditions (i.e., the maximum number of neutrons that can be found at the site without the direct

presence of hydrogen). This term is unique to each site and is found through the calibration process, explained in detail in Section 2.2.


The detection of background neutrons in the atmosphere, as a method to infer estimates of field scale soil moisture, was first described in Zreda et al., (2008). In that study, the authors demonstrated that neutron intensity above the surface was inversely correlated with the amount of moisture in the soil below. This was developed further in Desilets et al., (2010), where the initial form of equation 1 was first described and applications of this technology continued to be explored within the earth sciences

community (Desilets 2011; Franz et al., 2012; Rivera Villarreyes et al., 2011). A large-scale network of these sensors was subsequently deployed across the United States leading to the Cosmic-Ray Soil Moisture Observing System (COSMOS) (Zreda et al., 2012).

After the establishment of the first national-scale network in the US (Zreda et al., 2012), other countries such as Australia

(Hawdon et al., 2014), Germany (Zacharias et al., 2011, Bogena et al., 2016), and the UK (Evans et al., 2016, Cooper et al., 2021) established their individual national networks, as well as additional sensors located in smaller networks or individual sites. Sensors from these networks have in some cases been running for up to 10 years, which can be potentially valuable for the understanding of soil hydrology. As these networks have grown so has the literature surrounding best practices for calibration and correction of the sensor signals, allowing us to have a lower uncertainty in CRNS soil moisture estimates (Franz

et al., 2012; Rosolem et al., 2013; Hawdon et al., 2014; Baatz et al., 2015; Schrön et al., 2017).  As a consequence of improvements to the signal correction and sensor calibration, a divergence in methods is noticeable between different networks. Each network inevitably implements its own protocol when correcting the neutron signal to give soil moisture estimates leading to a less harmonized dataset among networks. This is in part due to the difficulties that would be encountered in quickly changing data processing pipelines within already established databases. The benefit of such structures is that live data is

available to stakeholders through online portals. Unfortunately, the interdependencies of a database does not lend itself to quick changes and so a post processing method could alleviate some of these issues.

This lack of a harmonized global dataset can ultimately lead to limitations in the global assessment of this technology from multiple CRNS networks. Discrepancies in processing methodology can leave questions around information obtained, and

uncertainty propagated, from analysis and comparison of sensors in different networks, such as whether soil moisture signals can be attributed solely to environmental differences or processing differences. By not necessarily following all recommended correction steps, the estimated soil moisture products from these sensors or even networks can be seen as sub-optimal, potentially undermining their true value. An example of the impact of evaluation sub-processed cosmic-ray soil moisture data from the US network against land surface models is presented by Dirmeyer et al. (2016). There is a consensus to follow certain

steps and guidelines which are not uniformly applied across all networks. Known corrections to account for changes in atmospheric pressure, neutron intensity, atmospheric water vapour and aboveground biomass are applied differently and on

occasion not at all on some networks which could lead to different estimates of soil moisture (Zreda et al., (2012), Hawdon et al., (2014), Evans et al., 2016). For example, Rosolem et al., (2013) demonstrated the influence on the neutron signal that occurs from changes in atmospheric water vapour over time. When comparing processed soil moisture estimates with, and without this additional signal correction, they demonstrated a difference of up to 0.1 $cm^3/cm^3$ at a site at Park Falls, USA. Additionally, Hawdon et al., (2014) demonstrated the different approaches available in correcting neutron counts for incoming cosmic-ray intensity and showed that there is a noticeable difference in neutron counts and ultimately soil moisture depending on the chosen method. Schrön et al., (2017) provided an improved approach to CRNS calibration demonstrating that their revised approach improves accuracy of soil moisture estimates. Using UK sites as an example, Schrön et al., (2017) found that the Root-Mean-Squared-Error (RMSE) of soil moisture estimates from the CRNS was reduced from 5.3% volumetric, using the conventional calibration approach, to 1.4% volumetric, using the revised calibration approach. Improvements in accuracy were identified at all the sites they analysed. Although this revised approach is being adopted in more recent studies (Cooper et al., 2021), this is not always the case (such as the original sites in the COSMOS network) and can mean that sites in different networks have been calibrated using different methods.

In order to mitigate this ongoing issue of lack of harmonization in the soil moisture estimates from the CRNS technology, we present here an open-source python tool to process raw CRNS data into soil moisture estimates, using the most current methods identified in the literature. It is designed to allow a user to apply consistent data processing methods across sensors that may be located in different networks. Section 2 will describe the structure of the tool along with the relevant correction and calibration methods. It will also describe the site metadata creation process which is an additional aspect to crspy that is built to facilitate data analysis of many sites. Section 3 will discuss the implications of differing processing methodologies on soil moisture estimates, as well as the benefits of creating detailed metadata for post processing analysis.

## 2 The crspy tool

The **C**osmic **R**ay neutron **S**ensor **Py**thon tool (crspy, pronounced "crispy"): is a tool written in Python3 that has been developed to facilitate the processing of the global networks of CRNS data in a uniform and harmonized way. It is available through an open-source repository and can be installed into a user's python environment. The tool is designed to allow the easy implementation of the most up to date correction factors and calibration processes to any CRNS site globally, ultimately allowing for any user to access a harmonized dataset. Although it is designed for multiple sites from varied networks, crspy is versatile enough to process a single site as well. It is being provided to help facilitate research in the CRNS community and is not intended to state whether one networks processing methods are superior to another. It is the authors' opinion, however, that it is important for the community to consider creation of a best practice, as this will allow comparison of sensor data around the world in the future. In addition, crspy is structurally designed to accommodate new corrections and processing steps

that may become available in the future in an easy manner. By being open source, crspy can also serve as a development and testing tool for any new understanding of the CRNS technology, as well as a teaching tool for the community.

Figure 1 is a visual representation of the processes within crspy that converts raw sensor data into corrected soil moisture estimates. Due to the varied nature of input data, such as when different networks label data differently, it is first necessary for a user to correctly format input data following crspy's naming convention (see Table A1 in Appendix). Additionally, to organise the various input and output datasets a specific working directory folder structure is necessary. This allows crspy to

135 automatically handle the numerous sources of data. After installing the package a user can build this folder structure easily with the function `crspy.initial(wd)` where `wd` is a string representing the working directory location.

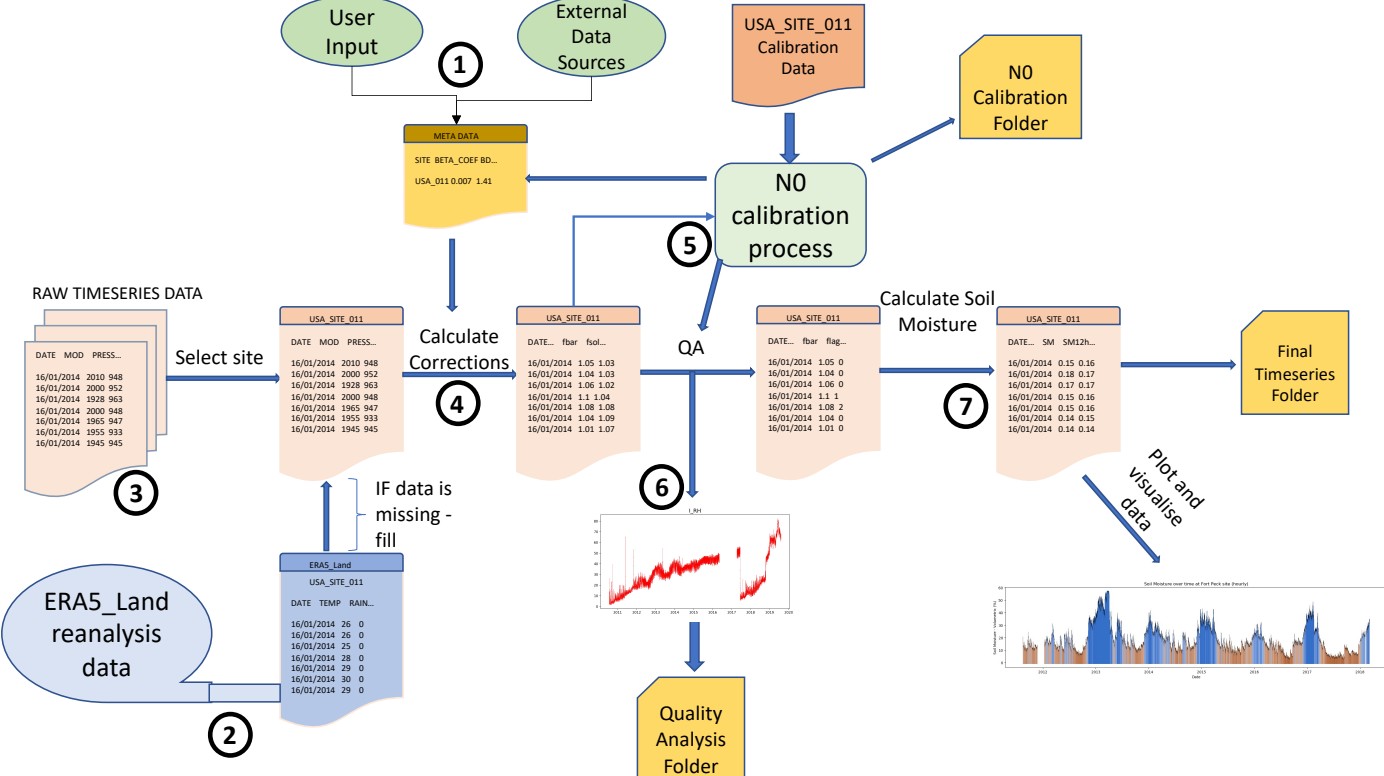

**Figure 1. The structure of crspy demonstrating all the modules that are used in creating soil moisture estimates. (1) represents the**
140 **metadata table which is a collection of site descriptors (e.g., soil texture, site elevation) (see section 2.4). (2), (3) and (4) corresponds to the ERA5-Land data collection and use for gap filling, data tidying and for the computation of correction factors (see Section 2.1). (5) represents the calibration process (if this option is selected) (see Section 2.2). (6) highlights the quality assessment steps undertaken (see Section 2.3). Finally, (7) represents the step where soil moisture estimates are calculated from the neutron counting rates (refer to equation 1).**

## 2.1 Data processing and correction

To obtain soil moisture estimates, we need to apply equation 1 at each time step in the data. The values will be obtained from time varying sensor data, external data products, static site-specific values and static values that are not site-specific. The coefficients $[a_0 \ a_1 \ a_2]$ are constants with values of 0.0808, 0.372 and 0.115, respectively, defined in Desilets et al., (2010). These values are fitting constants that describe the shape of the relationship between neutron counts and soil moisture, obtained from neutron particle physics modelling, and are the same for all sites. These values are stored in the `config.ini` file, which stores constant values for crspy.

### Site-specific soil properties

The site-specific soil parameters described in Equation 1 are $LW$, $WSOM$ (obtained from soil organic carbon) and $\rho_{bd}$. Due to the open data policies of many of the CRNS networks this data is usually available online (see data availability statement). These values should be defined prior to running the main crspy function and are stored and read from the metadata file.

The parameter $LW$ corresponds to the lattice water (%), which represents the hydrogen contained in the mineral structures of the soil (Hawdon et al., 2014). As fast neutrons are mitigated by hydrogen atoms, regardless of their source, this will have an overall impact on the neutron count rate. This value is usually obtained through analysis of soil samples taken from the footprint of the site sensor (Franz et al., 2012). The parameter $WSOM$ represents the water equivalent of soil organic matter (g/cm$^3$). Soil organic carbon (SOC) is obtained through analysis of soil samples and represents the total organic carbon in the soil at the site. Hawdon et al., (2014) discuss the need to convert this value into a water equivalent and provided a method for this (Equation 2). This is completed on the assumption that organic matter in the soil is cellulose which means that proportionally the water equivalent of this can be found by:

$$WSOM = SOC * 0.556 \tag{2}$$

The parameter $\rho_{bd}$ represents the dry soil bulk density (g/cm$^3$) and is a site-specific static value. It is obtained through analysis of soil samples and is used in the conversion of gravimetric soil moisture to volumetric soil moisture values. If dry soil bulk density data is unavailable for a site, crspy includes the option to obtain this value from the global data source SoilGridsv2 (see Section 2.4). In case of missing data, crspy takes advantages of built-in routines to fill out the information. In that case, if $\rho_{bd}$ or $SOC$ (used to calculate $WSOM$) are missing, then crspy will use the estimates collected from SoilGridsv2, which is collected in the metadata process. If $LW$ is unavailable a value of zero can be input into the metadata table by the user. There have been studies which demonstrate techniques to estimate $LW$ using soil clay content, which could be used to provide estimates that can be input to the metadata table (Avery et al., 2016, McJannet et al., 2017). Notice that the other site-specific

static value is the $N_0$ number. This number is found through the calibration process which is described in greater detail in section 2.2.

**Time varying values and correction methods**

The remaining values required to obtain $\theta_{vol}$ are $N_{raw}$ and $f_p$, $f_i$, $f_h$, $f_v$ which all vary with time. It is ultimately the relationship between $N_{raw}$ and $N_0$ which gives us our ability to estimate volumetric soil moisture once the additional corrections have been applied. The parameter $N_{raw}$ is obtained from the sensor data and will usually be representative of the number of neutrons counted over a one-hour time period. This is the measured raw (uncorrected) neutron count; however, we know that there are additional impacts on this count rate that require correction which are represented by the $f$ factors in equation 1. Changes in atmospheric pressure impact the neutron counting rate, the $f_p$ term corrects for this so that $N_{raw} * f_p$ gives the neutron count rate as if it were taken at the reference atmospheric pressure. Changes in incoming cosmic-ray intensity will directly influence neutron count rates as this is the source of fast neutrons and so the $f_i$ term will correct this to match a reference date in time. Atmospheric water vapour and above ground biomass are additional sources of hydrogen, outside of the soil moisture source we are interested in, and so the $f_h$ and $f_v$ terms adjust the count rate in consideration of this. These correction methods have been improved upon since the technologies first implementation with additional sources of uncertainty identified and equations designed to mitigate their impact.

There are occasional data availability issues observed at some sites. For example, meteorological variables are a necessary part of converting neutron counts to soil moisture estimates because they are needed to account for the numerous impacts on the signal, such as pressure corrections and atmospheric water vapour corrections. On occasion, some of the sites do not measure all the necessary variables considered to be essential to correct for additional sources on the neutron signal. External relative humidity sensors are essential in correcting for changes in atmospheric water vapour but are not always included in site data. When data is unavailable through in-situ site sensors, ERA5-Land (Muñoz Sabater, J., 2019) data are used to replace missing sensor data. ERA5-Land is a dataset, based upon the ERA5 reanalysis data, that combines modelled data with real world observations, resulting in a gridded, global hourly product at 9km resolution, provided publicly by the European Centre for Medium-Range Weather Forecasts (ECMWF). Previous iterations of the ERA reanalysis datasets (such as ERA-Interim) have proved useful by other global networks for the task of gap filling missing data, such as in the FLUXNET community (Vuichard and Papale, 2015). We are implementing a similar approach used by the FLUXNET community in crspy, and consequently to the global CRNS database, as we envision the potential of a merged database incorporating both flux tower and CRNS soil moisture data in the future. As the two measurement technologies show similar temporal and spatial footprints, their combined use can eventually lead to a better understanding of land-atmosphere interactions at the field scale, for example (Iwema et al., 2017). It is important to note that although the resolution is spatially coarser when compared with CRNS sites, the ERA5-Land dataset was chosen as a source for replacing missing sensors for three main reasons: (1) it covers the lifetime of all the CRNS

sites around the world which ensures all historical data to be used for gap-filling if necessary; (2) the dataset is produced at hourly resolution which matches the standard resolution of CRNS sites; (3) this is an open data source which aligns with our desire to develop a full open-source tool for CRNS data processing.

The ERA5-Land dataset includes key variables such as precipitation, temperature, and dewpoint temperature which can be used to correct for influence on the neutron signal, such as using dewpoint temperature when relative humidity sensors are not available at the site (Rosolem et al., 2013). Our choice also follows previous studies that demonstrated that ERA-Interim tended to perform best when compared with other global reanalysis products (Decker et al., 2012). ERA5, which ERA5-Land is derived from, has benefitted from a decade of research when compared to ERA-Interim and has been shown to be a great improvement (Hersbach et al., 2020).


**(i) Atmospheric Pressure correction ( $f_p$ )**

Changes in atmospheric pressure can have an impact on neutron counting rates measured by the CRNS (Zreda et al., 2012, Hawdon et al., 2014). This is attributed to the fact that higher atmospheric pressure reduces neutron counting rates as there are more particles in the air column that can slow fast neutrons down. In crspy this is corrected with the equation:


$$f_p = \exp\big(\beta(p - p0)\big) \tag{3}$$

where $f_p$ is the pressure correction factor (defined in Equation 1), $\beta$ is a coefficient to account for mass attenuation length at the site, $p$ is the atmospheric pressure at the site (hPa) and $p0$ is a reference atmospheric pressure (hPa) for the site, commonly

taken as the mean pressure for the site's elevation. The $\beta$ coefficient and the reference atmospheric pressure value are calculated for each location as a function of the latitude, elevation, and cut-off rigidity at the site as described in Desilets et al., (2021).

**(ii) Incoming High-Energy Neutron Intensity ( $f_i$ )**

It is important to correct for incoming neutron intensity as this will have a direct impact on neutron counting rates. Changes in the incoming cosmic-ray intensity will affect the number of fast neutrons in the atmosphere as increased cosmic-ray intensity will lead to an increased counting rate created through the cascade of reactions (Desilets et al., 2006). We use the data from the Neutron Monitoring Data Base (NMDB) available online, providing a collection of neutron monitoring sites from around the world. The NMDB provides neutron counting rates at hourly resolution from monitoring stations around the world, its data

is considered the official distribution from each site principal investigator. The correction method currently varies across networks. For example, the COSMOS (USA) originally corrected the data by comparing neutron intensity to a pre-defined

reference date, in that case, assumed to be 01 May 2011. The Jungfraujoch neutron monitoring station in Switzerland was used as a reference site. The calculation for this is shown as follows:

$$f_i' = \frac{I0}{Im} \qquad (4)$$

where $Im$ is the incoming cosmic-ray intensity at sensor measurement time and $I0$ is incoming neutron intensity at the decided reference date and $f_i'$ here is used to define this particular incoming cosmic-ray intensity correction factor, to avoid confusion with $f_i$ from equation 1.


The default approach in crspy, however, is to use the approach outlined in Hawdon et al., (2014) where the Jungfraujoch monitoring station is used but an additional correction for differences in site cut-off rigidity is applied with:

$$Rccorr = -0.075(Rc - Rcjung) + 1 \qquad (5)$$


where $Rccorr$ is the correction for differences in cut-off rigidity (GV), $Rc$ is the cut-off rigidity at the sensor location and $Rcjung$ is the cut-off rigidity at the Jungfraujoch monitoring station (which has a value of 4.49 GV). This is applied at each time step to give a final corrected value with:

$$f_i = (f_i' - 1)Rccorr + 1 \qquad (6)$$

Ultimately $f_i$ is the similar to $f_i'$ but contains an additional correction to account for the difference in cut-off rigidity between the CRNS site being processed and the Jungfraujoch neutron monitoring reference site.

The Australian CosmOz network uses a different approach which does not always use the Jungfraujoch as the reference monitoring station. Instead, this network will change the reference station based on the stations which has the closest cut-off rigidity (GV) to the sensor site from the Neutron Monitor Database (Hawdon et al., 2014). This option is also available in crspy when running the main processing function `crspy.process_raw_data(fileloc, intentype="nearestGV")` by invoking intentype with the "nearestGV" option. This involves identifying the NMDB

site with the nearest cut-off rigidity and applying equation 4.

**(iii) Atmospheric Water Vapour ( $f_h$ )**

Hydrogen atoms can slow down fast neutrons leading to a reduction in the count rate with increasing atmospheric water vapour. This signal needs to be removed to ensure that neutron counting rates are attributed to soil moisture and not moisture in the air. This is corrected at each time step with the following equation (Rosolem et al., 2013):

$$f_h = 1 + 0.0054 \times \rho v \tag{7}$$

where $f_h$ is the atmospheric water vapour correction factor and $\rho v$ is absolute humidity (g m$^{-3}$). Some sites do not have external relative humidity sensors that can be used to calculate vapour pressure, which can be used to calculate absolute humidity along with temperature. When this is the case then ERA-5 Land data can be utilised by converting dewpoint temperature (°C) to vapour pressure (kPA) (for further information on the steps to obtain absolute humidity from standard meteorological variables, please refer to the appendix section in Rosolem et al., 2013).

Arguably, ERA5-Land data presents a spatial mismatch with the cosmic-ray sensor, whilst also being a non-direct measurement of environmental variables. The majority of CRNS sites in the USA have not been deployed with a set of standard meteorological measurements, and only a few are co-located with external monitoring stations. Hence in this case, ERA5-Land data is critical to ensure neutron counts are appropriately corrected for water vapor variations at these sites. Our preliminary analysis suggests that correcting neutron counts with ERA5-Land data provide superior results compared to not applying the correction at all due to a lack of meteorological data (Figure 2). In this example, meteorological data at the ARM site in Oklahoma is available from a nearby flux tower (Biraud et al., 2021). Notice how the processed soil moisture timeseries corrected with ERA5-Land data closely follows the soil moisture estimates produced when using the in-situ meteorological data (Figure 2a). Neglecting this correction can lead to significant underestimation of soil moisture, especially during the wet seasons. Figure 2c helps to visualise these impacts by showing the difference in obtained soil moisture with correction using ERA5-Land data as well as with no correction applied, both compared against a soil moisture corrected with in-situ data.

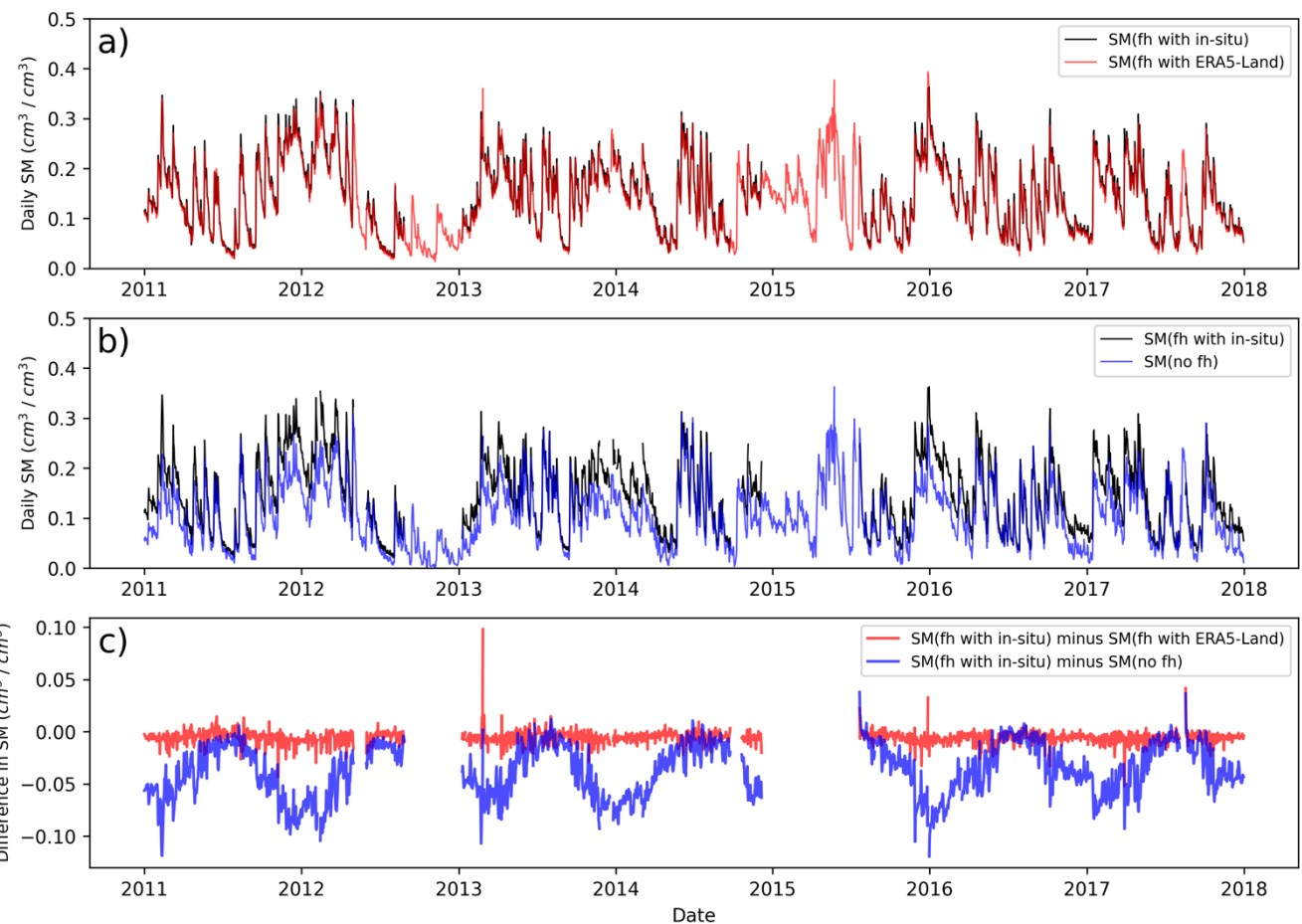

**Figure 2 shows the soil moisture (SM) record at the ARM-1 CRNS site in USA. Figures 2a and 2b show in black the SM product when corrected using in-situ data. The red line in 2a is SM product corrected with ERA5-Land data in place of temperature and relative humidity sensors. The blue line in 2b shows the SM product when not correcting for atmospheric water vapour (fh). Figure 2c shows the difference between the in-situ corrected SM and the alternative correction methods.**

**(iv) Above Ground Biomass (AGB) ($f_v$)**

Similar to other sources of hydrogen, biomass can also affect the neutron counting signal. There have been numerous attempts to identify the relationship between AGB and neutron count rates (e.g. Rivera Villarreyes et al., (2011), Baatz et al., (2015), Heidbüchel et al., (2016) and Tian et al., 2016). Unlike other sources of hydrogen, AGB is sometimes not available from local samples at each site. In order to reduce the impact of AGB on the measured neutron signal, crspy currently uses a static estimated value for each site from the European Space Agency (ESA) Climate Change Initiative (CCI) global dataset and apply the correction method based on the work of Baatz et al., (2015), who found a linear relationship between above ground biomass and neutron counting rates.

The following equation is used:


$$f_v = \frac{1}{1-(0.009*agb)}$$
(8)

where $f_v$ is the above ground biomass correction factor and $agb$ is the dry above ground biomass at the site (kg/m$^2$). The ESA CCI database provides above ground biomass estimates as a global gridded data product at 100m resolution (Santoro and
Cartus, 2019). As the ESA CCI data currently used is a static value in time, it will not impact the soil moisture estimates, in principle, because the correction is applied on both the $N_{raw}$ and $N_0$ numbers, hence mitigating any impact. Nevertheless, we have included this routine in crspy in this form as we anticipate improvements to dynamical above ground biomass corrections in the future, at which point crspy can be updated to include the latest theory that can be applied across all sites (Franz et al., 2018; Vather et al., 2020; Fersch et al., 2020). Further improvements to be able to dynamically account for biomass changes
at all CRNS sites will be important for reliable estimation of soil moisture dynamics, especially when analysing sites with land-use changes or cropping cycles.

## 2.2 Sensor calibration

The above steps give us all the values in equation 1 necessary to provide a soil moisture estimate, except for $N_0$. A required step in processing, and eventually using the data, is to calibrate the CRNS to the specific conditions found at the site of interest.
Without this step, the soil moisture can potentially have significant biases and deemed unusable. Alternatively, the uncalibrated measurement can only give you a rough idea about the dynamics of the soil wetness conditions in relative terms. The calibration step typically requires multiple soil samples (typically 100s) taken from within the sensor footprint and oven dried to get an accurate representation of soil moisture at the calibration time. These samples are then weighted and averaged to give a field scale soil moisture estimate of the sensor footprint (note that we use dry soil bulk density, $\rho_{bd}$, sampled within the footprint to
estimate volumetric water content in cm$^3$ cm$^{-3}$). The crspy tool uses the soil moisture averaging method obtained from field samples proposed by Schrön et al (2017), which is based upon the original work of Köhli et al., (2015). The method provides an updated approach for weighting soil moisture samples taken within the footprint that considers spatial distance from the sensor of each sample as well as the influences of pressure and humidity during the sampling period. This allows for a more accurate estimate of independent soil moisture within the CRNS footprint for the calibration step. Schrön et al. (2017)
suggested improved sampling strategies which included samples closer to the sensor (< 5m radius from the sensor) and sample locations guided by knowledge of local hydrological features. The data required for the calibration step includes the date of the sample, an integer to represent each soil moisture profile (a core of soil taken from within the sensor footprint), the depth of each sample within each profile, the distance from the sensor, and the volumetric soil moisture of the sample. Again, these should be named following the template requirements by crspy (see Table A2 in Appendix). Calibration datasets are openly
available from some of the networks at already established sites, such as CosmOz and COSMOS, and can be obtained from

their respective websites. Alternatively, if a user was setting up their own sensor, then a sampling campaign would be required such as that described in Schrön et al., (2017).

With regards to number of calibration days, crspy is flexible enough to process both single-day or multiple-day calibration
campaigns. Multiple calibration campaigns were shown to improve the CRNS signal (Iweema et al., 2015). For the case of multiple-day calibration, all calibration days should be presented in a single table, ensuring that the correct dates of each sample period are provided, and following the same formatting and naming requirements used for single-day calibration.

Finally, when running crspy for a single site, the user is able to turn on or off the calibration process. This is included because calibration only needs to be done once, as $N_0$ does not vary with time. When the calibration step is turned on, crspy will call
the calibration routine and write the output to the metadata table in the column 'N0'. If the calibration routine is turned off, crspy will skip this step and simply read the $N_0$ number for the site from the metadata. Alternatively, the user can provide the $N_0$ coefficient independently in the metadata table and skip the calibration step completely by always having it off in crspy.

## 2.3 Quality assessment

All data should be checked for quality to ensure that erroneous data is not included, and crspy includes some automated steps to begin this process. All networks implement quality assessment on neutron counts in order to remove poor quality data (e.g., Zreda et al., 2012; Hawdon et al., 2014; Evans et al., 2015). In crspy, we remove suspicious data points by applying flags to neutron counts that fall within four categories, the below rules are consistent with the application in other networks:

1.    Counts that differ by 20% from the previous time step are removed
2.    Counts below 30% of N0 are removed
3.    Counts above N0 are removed
4.    Battery voltages below 10V are removed

Flag 1 is applied on the raw, uncorrected neutron count as we are interested in identifying sudden jumps in counting rate in the sensor that a believed to be in error. Flags 2 and 3 are applied to the corrected neutron count. This is because the N0 number is itself a corrected number (i.e. it is the maximum number of neutrons at the site under theoretical dry conditions, once additional environmental influences on N have been taken into account and removed from the signal). In the case of flags 2 and 3 the N and N0 number need to both be corrected to be comparable.

Additionally, crspy will output time series diagnostic plots of all variables used for identifying patterns in data that point towards potential issues which may require a small subset of the data to be removed manually (this, of course, depends on the quality of the data from individual sites and, therefore, cannot be fully automated).

 **2.4 Metadata**

Metadata is an important piece of information that allows the user to better describe each site characteristics beyond its soil moisture dynamics. The information can be extremely useful especially when multi-site regional to global CRNS stations are to be analysed simultaneously. The metadata of each site is stored in a tabular format within the folder structure of the working directory, a full description of the columns is given in the Appendix (Table A3 in Appendix). It serves two main purposes.
Firstly, it stores static site-specific variables that are used in computing estimated soil moisture values (e.g., $LW$, $SOC$ and $\rho_{bd}$). To provide an example, $\rho_{bd}$ is necessary to convert gravimetric soil moisture estimates into volumetric soil moisture estimates in equation 1. The $\rho_{bd}$ value is collected during the calibration campaign at each site and will vary between sites. It represents an averaged value taken from the soil samples and it is stored in the metadata. The user should also give each site a country code which represents the country it is located in and a unique site number for each CRNS site. The country code is
used to help identify geographic locations in analysis and helps when the site numbering of networks may overlap. Raw time series data should be titled with the country code and number in the following format: $country$_SITE_$sitenum$.txt. Where $country$ is a capitalised letter code and the $sitenum$ is a 3-digit number. For example, sensor data for a site in the UK could be titled: UK_SITE_101.txt. This $sitecode$ (i.e., UK_SITE_101) is used to identify each site when organising the outputs, as well as a lookup code for constant variable values stored in the metadata.


A second purpose of the metadata is to act as a resource when analysing many sites together. The ability to classify catchments by physical characteristics can allow us to understand key similarities and differences between sites, an important direction in hydrological research (Wagener et al., 2007). To increase the value of the metadata, as well as including data collected at the site, global data products have been integrated. These products are all public products that a user can download and store
within the folder structure of the working directory. We realize that these global datasets are not a direct replacement for the invaluable information obtained at the site; however, in many cases, such pieces of information are not available, undermining any multi-site analysis. We believe the use of the datasets described in detail below can provide us key information at regional and global level. In crspy, a simple function is used to extract the information from the data products below when provided with the location of the CRNS (i.e., latitude and longitude):


(i) **ESA CCI Land Cover and Above Ground Biomass data:** The European Space Agency (ESA) Climate Change Initiative (CCI) provides numerous global data products that are useful in the earth sciences community. Land cover data and above ground biomass data are obtained from ESA CCI and stored in metadata for each site for analysis through identifying site differences and similarities. Both products are spatial consistent with the CRNS
sensor range (100m-300m) and are available globally. The usefulness of ESA CCI datasets in land surface modelling continues to be established (Li et al., 2017).

(ii) **International Soil Reference and Information Centre (ISRIC):** The ISRIC provides a global data product that gives estimates of soil properties on a 250m resolution grid. This is available as SoilGridv2 which is an updated (as of May 2020) iteration of the original SoilGrid product (Hengl et al,. 2017). The properties are estimated from collections of ground measurements that are compiled by the World Soil Information Service (WoSIS). WoSIS provide standardised soil profile data to facilitate the creation of products such as SoilGrid (Batjes et al., 2020).

(iii) **ERA5-Land:** As discussed previously meteorological variables from ERA5-Land data can be downloaded for each site. Mean annual precipitation and temperature data is stored, along with derived Köppen-Geiger classifications.

## 2.5 Running the tool

Once the working environment has been prepared the data can be processed with `crspy.process_raw_data(fileloc, calibrate=True, intentype=None)`. Where fileloc is the location of the raw sensor data, the calibration process can be turned on or off as a Boolean descriptor and intentype can be left as `None` to enact the default process for incoming neutron intensity correction or can be changed to "`nearestGV`" to utilise the alternative method. Once applied crspy will process the raw data using the provided information to give soil moisture estimates and will output figures and tables into the folder structure of the working directory. A description of the final output file and what each of the standard columns represent is given in Table A4.

## 3 Discussion

### 3.1 Benefits of data harmonization

As mentioned previously, one of the key purposes of crspy is the easy and harmonized processing of CRNS sites from around the globe, as there is currently no true consensus on what correction steps are implemented in different national networks. These technical differences can lead to changes in outputs which may result in non-optimal conditions for regional/global analysis from multiple countries. Whereas some users may wish to understand changes at one particular site, inter-site comparisons are limited when each site could be processed in a different way. In this section, we highlight such impacts with one example related to the individual sensor corrections steps, and their impact on the final soil moisture estimates.

Table 1 outlines three identified methods that are currently employed across different networks. Method $p\_int1$ is the method employed at the COSMOS (USA) network, which lacks the atmospheric water vapour correction and applies an intensity correction using only the Jungfraujoch neutron monitoring site directly. Method $p\_int2\_awv$ closely resembles the CosmOz (Australia) network methodology, which does apply the atmospheric water vapour corrections and an intensity correction that differs from method $p\_int$. In this case, the neutron monitoring station used as an incoming neutron intensity reference is

changed to the nearest station with a similar cut-off rigidity to the CRNS site being corrected. Method $p\_int3\_awv\_agb$ is the default crspy method, and resembles the methods used by COSMOS-UK, while also allowing for the above ground biomass correction to the neutron signal. In this final case, the intensity correction uses Jungfraujoch as its reference site but with an additional correction to account for differences in cut-off rigidity between Jungfraujoch and the site (equation 5).

**Table 1. The three identified methods of correcting neutron signals in use.**

|  | Method $p\_int1$ | Method $p\_int2\_awv$ | Method $p\_int3\_awv\_agb$ |
|---|---|---|---|
| Atmospheric pressure correction | Yes | Yes | Yes |
| Incoming neutron intensity correction | Jungfraujoch NMDB (no GV correction) | Nearest GV NMDB (variable locations) (Hawdon et al., (2014)) | Jungfraujoch NMDB **plus** additional correction for site GV (see equation (5) and Hawdon et al., 2014) |
| Atmospheric water vapour correction | None | Yes (Rosolem et al., (2013)) | Yes (Rosolem et al., (2013)) |
| Above ground biomass correction | None | None | Yes (Baatz et al., (2015)) |

With all these different correction approaches applied independently from each national network, we investigate both the impact on the measured neutron counts and consequently the propagated effects on the estimation of soil moisture. Figure 3 shows two sites with distinct hydroclimatic regimes, both taken from the COSMOS-USA network, that have been processed in the three identified methods (see highlighted star markers in Fig. 4 and Fig. 5). The Santa Rita Creosote site is shrubland dominated with a soil categorized predominantly by sandy loam type located in Arizona, USA. The site has a mean annual temperature of 19 °C and a mean annual precipitation of 335 mm, which primarily falls in winter storms and monsoonal summers (Köppen-Geiger climate classification BSh, a hot semi-arid climate). Climate data taken from ERA5-Land and Köppen-Geiger classification derived from ERA5-Land data using the method outlined in Peel et al., (2007). The Wind River site is an old-growth mixed conifer forest site in Washington, USA. The site is much wetter than the Santa Rita Creosote site, with an annual precipitation of 2,200 mm, and much colder, with an average annual temperature of 8 °C. Precipitation at Wind River tends to fall all year round but with slightly lower rates in the summer period (Köppen-Geiger classification is Csb, a Mediterranean climate mild with dry, warm summers). Climate data has been extracted from ERA5-Land and Köppen-Geiger classification derived from 10 years of ERA5-Land data using the method outlined in Peel et al., (2007). The raw neutron data from both sites were obtained directly from the COSMOS network, representing the $p\_int$ case in Table 1. In addition, in order to compare the impact of the different correction approaches outlined in Table 1, the raw data from the CRNS at both sites, have been processed in crspy to give the corrected signals for methods $p\_int2\_awv$ and $p\_int3\_awv\_agb$.

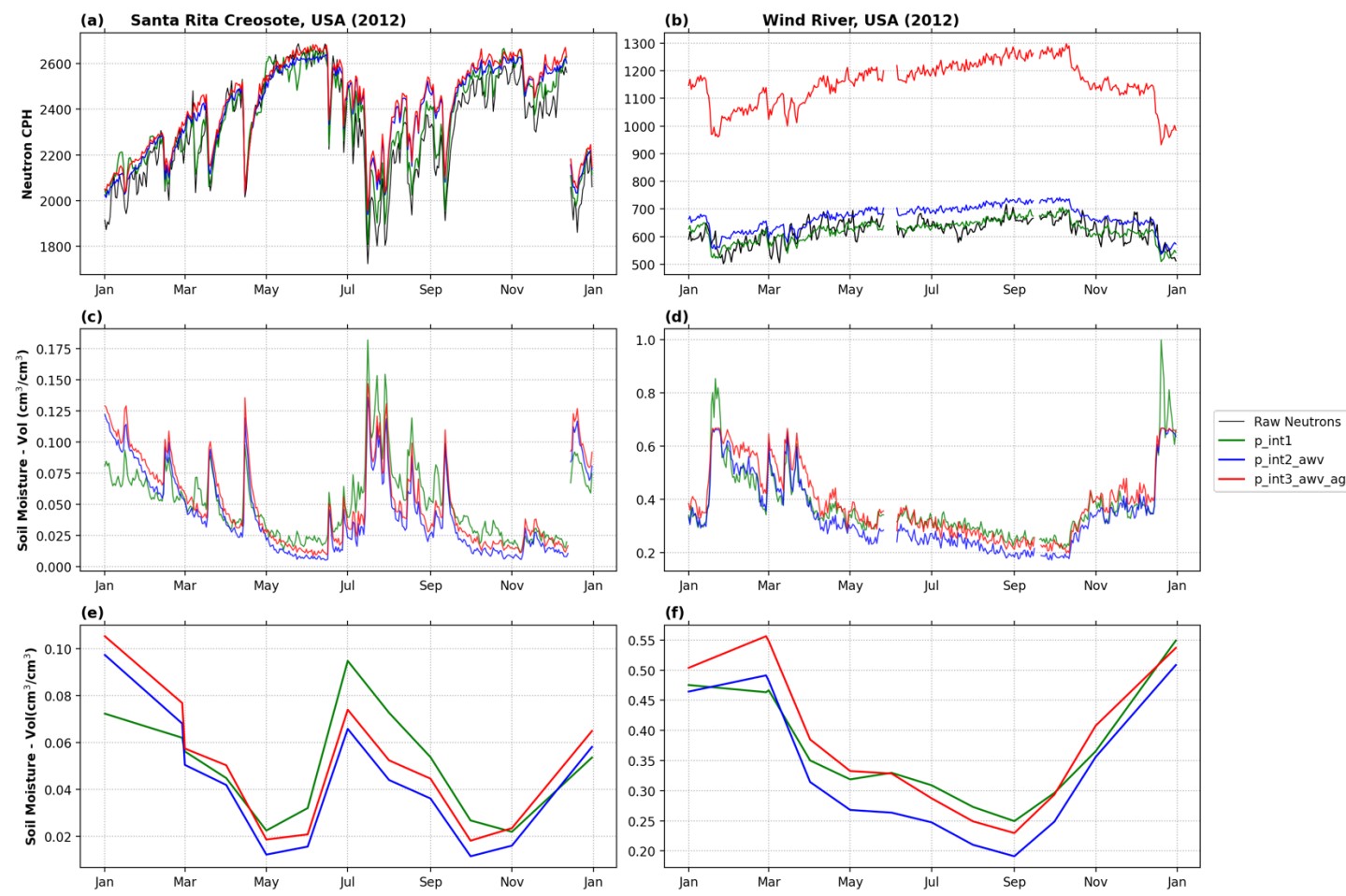

**Figure 3. Example of CRNS data obtained at two distinct sites: Santa Rita Creosote (a, c, and e) and Wind River (b, d, and f). Daily neutron counting rates (raw and corrected based on the different strategies outlined in Table 1) are shown in panels (a) and (b). Derived soil moisture estimates (cm$^3$ cm$^{-3}$) are shown at daily and monthly timescales in panels (c) and (d) and panels (e) and (f),**
**respectively.**

It is clear to see the inverse relationship between neutron count rates and soil moisture, most noticeably at Santa Rita Creosote (Figures 3a and 3c). The soil moisture here tends to be low, such as in June when it was below 0.05 cm$^3$ cm$^{-3}$, which is to be expected in a hot semi-arid environment. Sudden spikes in soil moisture can be attributed to precipitation events, with the summer monsoonal precipitation causing a sudden increase in the mean soil moisture values for the months of July, August,
and September (and, inversely, periods corresponding to decreases neutron counting rates). It is also clear that the method chosen has an impact on soil moisture values. This is most notable when comparing the $p\_int1$ method with both the $p\_int2\_awv$ and $p\_int3\_awv\_agb$ methods. During the summer months, the $p\_int1$ method appears to estimate higher soil moisture values compared to the other two methods (both appearing to be much more closely aligned to each other). This is likely due to the fact that the $p\_int1$ method does not account for changes in atmospheric water vapour. As a consequence,

during the monsoonal summers when there is more hydrogen in the atmosphere from increased humidity, the relatively high water vapor in the atmosphere is incorrectly attributed to additional soil moisture. This is because the CRNS records wrongly attribute the decrease (attenuation) of neutron counts due to water vapor to an increase in soil moisture, causing an over estimation. For example, even early in March there is a sudden rise in soil moisture from the $p\_int1$ estimates which does not appear in the other two methods (Figure 3c). This suggests that rather than a sudden rise in soil moisture, this was actually a rise in atmospheric water vapour. This demonstrates the importance of removing external impacts on the neutron signal as they could be incorrectly attributed to soil moisture dynamics. The negative effect of neglecting such correction, for example, can be pronounced even more on monthly estimates of soil moisture due to the aggregated nature of this error (Figure 3e).

The Wind River site is a much wetter site when compared to Santa Rita with its driest month matching Santa Rita Creosotes wettest month. It is worth noting in that in the case of Wind River there is a much larger difference between the neutron count rate of method $p\_int3\_awv\_agb$ compared to the other methods (Figure 3b). This is because the $p\_int3\_awv\_agb$ method includes an above ground biomass correction, using the ESA CCI above ground biomass product to calculate a correction. Currently, as this correction is applied using a static aboveground biomass value (constant with time), the impact of the correction is not translated to differences in estimated soil moisture. This is due to the correction being applied to both the neutron counting rate and the $N_0$ term as well. With dynamic data, that represents changes in above ground biomass over time, we would be able to improve our estimates of soil moisture, as the impact of changing above ground biomass could be removed from the neutron signal. One additional noticeable feature that crspy implements is the capping of soil moisture to more realistic values, in this case 0.68 cm$^3$ cm$^{-3}$. The $p\_int1$ method does not do this and so there are physically impossible values of volumetric soil moisture in February and December, as seen in figure 3d. In crspy maximum values for soil moisture are estimated by inferring porosity of the soil:

$$sm\_max = 1 - \left(\frac{\rho_{bd}}{density}\right) \tag{9}$$

where $sm\_max$ is the maximum volumetric soil moisture value (cm$^3$ cm$^{-3}$), $\rho_{bd}$ is soil bulk density (g/cm$^3$) and $density$ is the density of ground material (estimated with an assumed density of quartz at 2.65 g/cm$^3$). If a user did not wish to enable this cut-off value, then the value for $sm\_max$ can be set to one in the metadata.

At the Wind River site, the differences between $p\_int2\_awv$ and $p\_int3\_awv\_agb$ are much more noticeable, especially when the soil moisture estimates are aggregated to monthly timescales (Figure 3f). This observed difference is due to the fact that these methods do not apply the same correction for incoming cosmic-ray intensity ($f_i$ ). Such differences are caused by the choice of correction rather than physical controls on soil water dynamics. This can lead to inaccurate comparisons across sites from different national/regional networks. For example, identifying useful soil moisture signals that can be used to categorise

the hydrology of sites will be an important tool for understanding differences and similarities with regards to hydrology. Branger and McMillan (2020) demonstrated this in their paper looking to identify useful soil moisture signals that can be

robust, discriminatory, and representative, with research continuing in this area of developing useful diagnostic soil moisture signatures (Araki and McMillan, 2020). When reducing large time series data into signatures, such differences can be aggregated which could begin to affect conclusions. However, the authors stress here that it is not within the scope of this work, nor the intention, to identify which method is better or worse than the other, but rather highlight the potential negative impacts of the lack of a harmonized dataset for large-scale global assessment of the CRNS technology.


## 3.2 Usefulness of crspy metadata

Metadata can be used to describe the network of CRNSs around the world geographically, climatologically, and hydrologically. To achieve this, crspy compiles relevant data obtained directly from the sensor, key data descriptors provided from each site or network, and from global data products. Wagener et al. (2021) discuss the need for high quality metadata to improve our

ability to understand the knowledge accumulation in the field of hydrology. The metadata can be valuable in separating relevant sites in different groups, for example researchers may be interested in understanding how soil moisture behaves at sites above 2000m elevation with certain land use types and given particular weather events (Chen et al., 2020); or sites where mean annual precipitation is above/below a certain threshold; or even grouping sites by different land cover or soil types. So called meta-analyses can help a researcher identify which sites should be included in their studies and which can be excluded (Evaristo

and McDonnell, 2017).  The metadata provided by crspy allows the user to quickly obtain any grouping of interest in an easy and accessible way.

In order to demonstrate some of the features that can be easily accessed with the help of metadata, we show an example using the compiled COSMOS network data for continental USA (CONUS). Some of this data are taken directly from the network

website and then processed using the `crspy.fill_metadata()` function. This function collects information from global data products at specific site location (i.e., latitude and longitude), as well as using meteorological data from ERA5 Land to produce annual meteorological summaries (e.g., mean annual temperature, mean annual precipitation, Köppen-Geiger climate classification, etc). Figure 4 gives an example of how the metadata can be easily used to show the location of each sensor in the continental USA domain (CONUS) based upon the supplied with additional information, in this case, main land cover

classes obtained from the CCI ESA Land Use data. An important step here is that the user is not required to download and process the land cover data separately and individually. crspy incorporates that step for the user seamlessly.

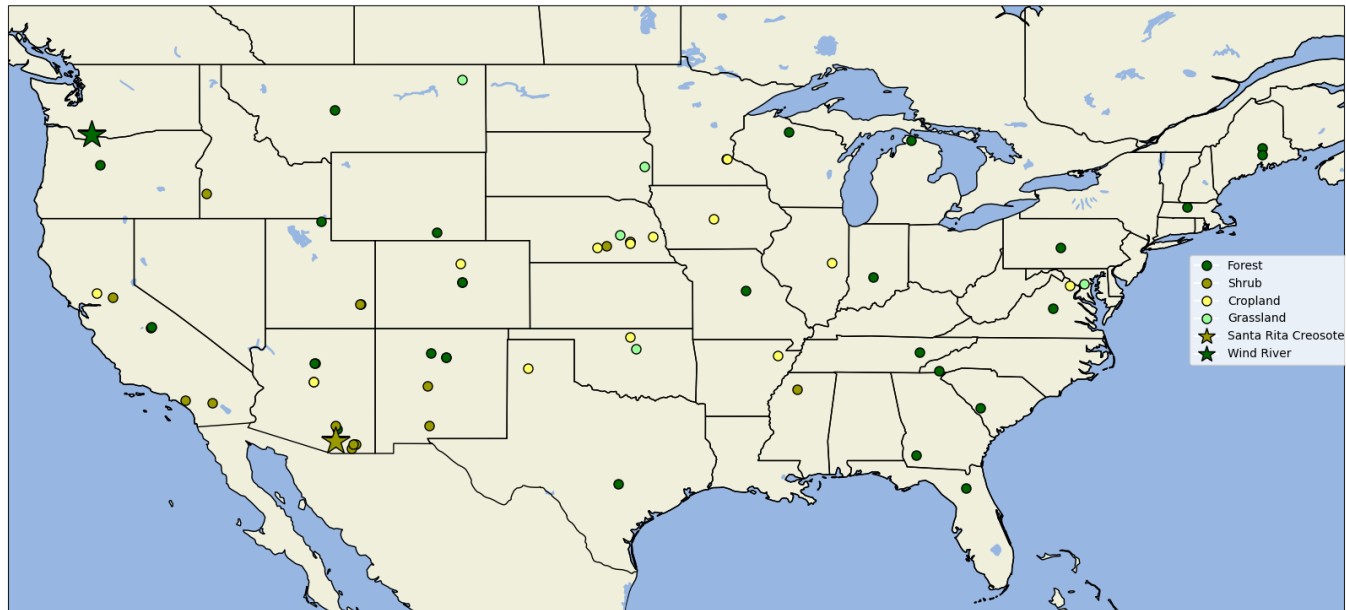

**Figure 4. Map showing the location of CRNS sites from the COSMOS network across CONUS. The colours are representative of the land cover types obtained from the ESA CCI global database and the stars highlight the location of the two sites processed above (i.e., Santa Rita Creosote and Wind River).**

In addition to locating the CRNS stations and identified the main land cover type, Figure 5 shows a scatter-histogram of the sites across CONUS providing additional annual meteorological summaries, namely mean annual temperature and mean annual precipitation. The scatterplot still retains the information about the main land cover type obtained from the ESA CCI global database. In addition, both meteorological variables are shown as side-histograms and were computed using the ERA5 Land data. Initial analysis indicates that CRNS classified as shrublands tend to be relatively warmer and drier. Grassland and forests tend to be wetter while showing a wider range of temperatures. Croplands are slightly warmer than grassland and forests but still showing lower temperatures than those observed in shrublands. However, croplands also indicate a slightly wider range of wetness, when compared to the grassland and forest sites, as observed from the total annual precipitation. This could be useful when deciding which sites should be used in a particular study, such as a study on soil moisture dynamics in shrublands with low overall precipitation. Alternatively, it can be used in big data analytics when trying to identify the dominant mechanisms on soil moisture dynamics globally.

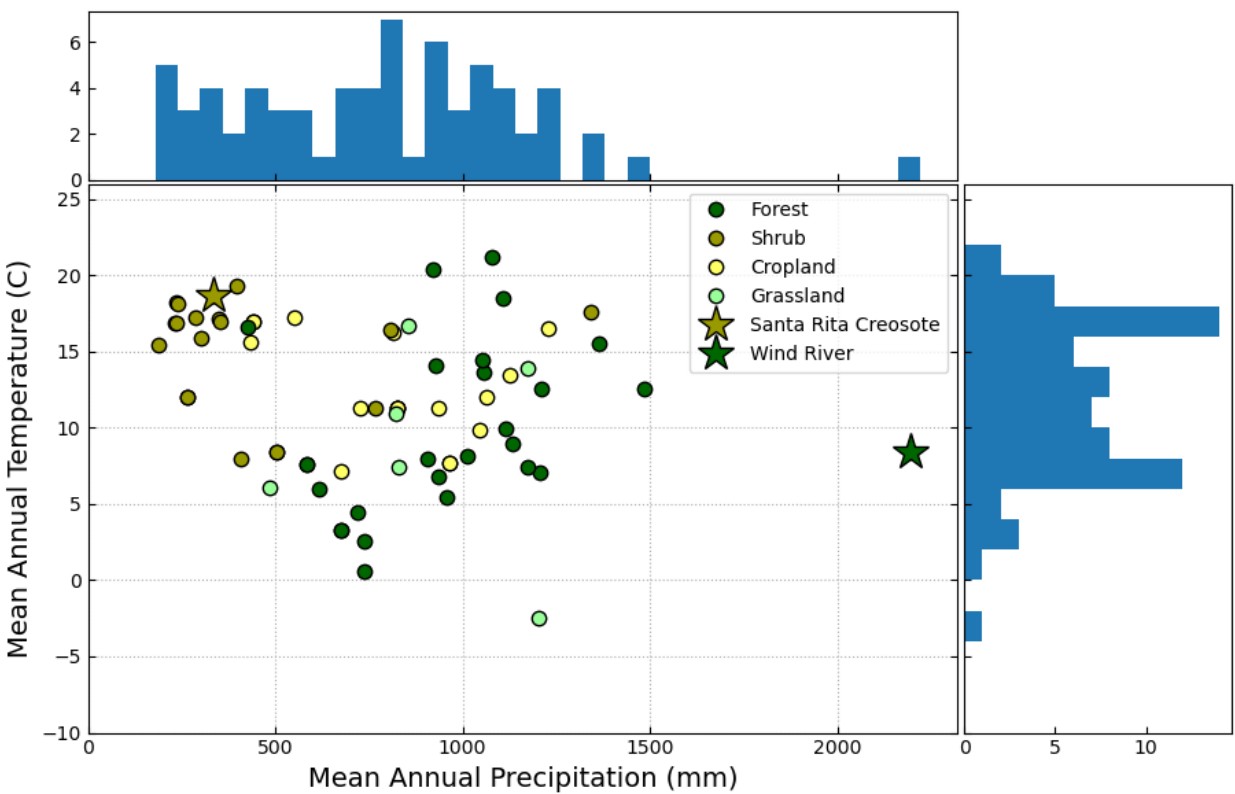

**Figure 5. Scatter-histogram showing the Continental USA CRNS sites and some of their climatological characteristics. The units for the histograms are number of sites for each bin. The colours represent land use types which has been identified from ESA CCI Land Use global data set. The stars highlight the location of the two sites processed above (i.e., Santa Rita Creosote and Wind River).**

The objective of metadata in crspy is to easily collect a wide range of site characteristics information that can be used to improve our knowledge of soil moisture, and consequently, other hydrological and environmental processes beyond just a single site. By doing so, it allows for knowledge accumulation across multiple sites (from local to regional and even global) highlighting key similarities and any emergent patterns (e.g., hydroclimatic, ecological, etc). Metadata analysis has not yet been fully exploited in hydrological sciences (Evaristo and McDonnell, 2017) but can also contribute to knowledge

accumulation which can be translated to help the designing of improved or new perceptual or conceptual models (Wagener et al., 2021). An early example of that within the cosmic-ray neutron sensing community is clearly demonstrated by Shuttleworth et al. (2013) during the conceptual development of the COsmic-ray Soil Moisture Interaction Code (COSMIC). COSMIC was developed as a forward observational operator allowing for the conversion of simulated soil moisture profile by land surface or hydrological models into equivalent neutron counting rates which facilitates data assimilation applications (Rosolem et

al., 2014). By collecting and accumulating information from (at that time) 42 available COSMOS sites (see Table 1 in Shuttleworth et al., 2013), the authors were able to simplify the requirement for two of the prescribed parameters by establishing relationship with dry soil bulk density (see Figure 6 and Equations in 6 and 7 in Shuttleworth et al., 2013). crspy will certainly facilitate such efforts in the future to help both experimental and modelling scientists with the potential to reach other disciplines beyond traditional hydrological and environmental sciences. For example, a prototype

version of crspy has already used for Space Weather application recently (Hands et al., 2021).

## 4. Future direction

In this paper we have presented crspy, an open-source python tool for the processing of cosmic-ray neutron sensors. Our aim

in developing crspy is to provide a tool to the community that can provide methods to process CRNS data easily and that can be updated in the future to keep pace with our increasing understanding of the sensor signal. Due to this evolving understanding of the sensor, we expect to be updating crspy regularly in the future to accommodate our new understanding of the technology along the years.

Köhli et al., (2021) recently presented research that demonstrates a revised formulation of the key equation that converts neutron counts into soil moisture estimates (see Equation 1). This emphasises the need to be able to update CRNS estimates to keep pace with the research as well as to test newer formulations across a range of sites quickly. In version 1.2 of crspy we maintain the Desilets et al., (2010) version of equation 1 as the default setting but have provided a supplementary document that describes how a user could update crspy on their home machine to implement the revised approach. This document serves

two functions, it is to demonstrate how to update crspy so that researchers may be able to test newer methods on a broad range of sites. But it also speaks of a more general need to agree on what is the standard approach in processing CRNS data. We believe it will be an important step in the future for the numerous stakeholders in CRNS measurements to agree upon a standard approach. This must be decided as a community and we should look towards the positive steps other communities have taken in this regard, such as the flux community (Novick et al., 2017).


Another aspect of development in crspy will be in making it more accessible and user friendly. We consider one of the key functions of crspy is to act as a tool for researchers, giving a way to update processing methods and apply it quickly on a collection of data. On top of this we would hope it can be used as an education tool, helping newer users understand how the sensor functions and what is required to fully correct it based on our current understanding. This could include developing

crspy into frameworks such as Python Dash, which are powerful tools for exploring data.

## 5. Summary

Soil moisture is an important component of the hydrological cycle and understanding its dynamics at relevant spatiotemporal scales is critical especially with recent advances of global land surface and hydrological models. The CRNS technology is able to provide estimates of soil moisture at the sub-kilometre scale and at hourly resolution. This is particularly relevant now as we continue to move towards hyper-resolution global modelling efforts. Over the years, with an increase of adoption to the technology, the CRNS community has acquired a better understanding about the benefits and limitations of this relatively novel technique. However, due to a lack of data harmonization across networks, undertaking global scale analyses is currently very limited and unexploited. Here, we introduced the crspy python package with the aim to facilitate users to process data easily with the most current methods, and most importantly, in a harmonized fashion. crspy is an open-source tool aimed to integrate the latest developed methodologies for CRNS to be used both for research and teaching activities. It integrates high-quality global data products (such as ERA5-Land) to ensure that all sites can be included in analysis. This is done in a similar way to other well-established global environmental networks such as the Ameriflux and Fluxnet.

Our examples of application demonstrated that when CRNS data is processed with different methodologies, it can ultimately lead to divergences in soil moisture estimates. This could potentially have a negative impact on the analysis and overall findings, especially when sites across multiple networks are evaluated simultaneously. By harmonizing data processes, we envisage that CRNS data will be used more widely by the global modelling and experimental communities, leading to further adoption of the technology. The objective of crspy is to provide an open and easy-to-use data processing platform that can enable easy processing of CRNS data. Additionally, crspy data collection relies on the production of an extensive metadata archive. This archive can be shared and used within the community to better understand key aspects about soil moisture from typical sampling locations, to inform on signature behaviour by different groupings. Crspy has been developed to show the potential for easily and efficiently processing CRNS data in a harmonized way. The aim is to promote the usefulness of free and open access data and engage the CRNS and research communities in the continued improvement of this product in the coming years.

## 6 Appendix

The appendix section consists of two appendices. Appendix A consists of four tables that outline the naming conventions required for crspy to run as well as the output table along with a description of each variable. When labelling input data, columns titles should match the styles below in the 'COLUMN NAME' column. This initial step will then allow crspy to run smoothly, as it uses column titles to identify relevant data sources. Appendix B provides some examples of the automatically generated outputs of crspy along with a description of their purpose.

**Table A1. The naming convention for CRNS input data. Networks can occasionally have different naming conventions (e.g., temperature is referred to as t1). Changing the column titles to the following format will allow crspy to function correctly.**

| COLUMN NAME | UNITS | DESCRIPTION |
|---|---|---|
| TIME | datetime | Datetime of the observation in UTC time. Format: "yyyy-mm-dd hh:mm:ss" |
| MOD | count | Moderated neutron count for time interval – the sensor tube is surrounded by a high-density polyethylene shield to remove thermal neutrons from the count rate |
| UNMOD | count | Unmoderated neutron count for time interval – a bare tube without the shield which will include thermal neutrons in the count |
| PRESS1 | hPa | Pressure sensor number 1. Usually the older analogue version that is somewhat less accurate |
| PRESS2 | hPa | Pressure sensor number 2. This will be used primarily and if unavailable PRESS1 will be used in place. |
| I_TEM | Celsius | Internal Temperature of the sensor box |
| I_RH | % | Relative humidity inside the sensor box |
| BATT | Voltage | Voltage of the battery |
| E_TEM | Celsius | External Temperature at the site. This would be an external reading. If not available then ERA5-Land data is used |
| E_RH | % | External Relative Humidity at the site. If not available then dewpoint temperature is used to find absolute humidity |
| RAIN | mm | Rainfall at the site. If local is available this is used - if not available it is obtained from ERA5-Land data |


**Table A2. The naming convention for the calibration data. This format should be followed and will allow the calibration module to be utilised.**

| COLUMN NAME | UNITS | DESCRIPTION |
| --- | --- | --- |
| DATE | Date. Format "dd/mm/yyy" | Date that the data was collected from the site |
| PROFILE | int | Integer to differentiate profiles. A profile being a single core. The core could then have multiple "layers". |
| LOC_rad | meters | Distance from the sensor for each sample in meters. |
| DEPTH_AVG | cm | The depth of the soil sample for each layer. Taken as the mid point of the layer. |
| SWV | % | The volumetric soil moisture of the sample. Should be given as decimal (i.e. 0.3). If it is given as a numeric percent (e.g. 30%) crspy will attempt to identify this and convert to decimal |





**Table A3. The naming convention of the metadata table.**

| COLUMN NAME | UNITS | DESCRIPTION | Required at start? |
|---|---|---|---|
| COUNTRY | - | Country code for location of site e.g. "USA" | Yes |
| SITENUM | - | Assigned 3 digit number for site: e.g. 001 | Yes |
| INSTALL_DATE | - | Date of site installation | No |
| LONGITUDE | degrees | Longitude of site | Yes |
| LATITUDE | degrees | Latitude of site | Yes |
| ELEV | m | Elevation above sea level of site | Yes |
| TIMEZONE | - | Time zone of the site | No |
| GV | gv | Cut-off Rigidity of site | Yes |
| LW | % | Lattice Water from site specific calibration data | Yes |
| SOC | % | Soil Organic Carbon from site specific calibration data | Yes |
| BD | g/cm$^3$ | Bulk Density from site specific calibration data | Yes |
| N0 | - | Theoretic maximum neutron count for site (dry conditions), calculated in tool and written. | No |
| AGBWEIGHT | kg/m2 | Live woody above ground biomass estimates from ESA CCI biomass data | No |
| RAIN_DATA_SOURCE | - | Declaration of the source of rain data. Currently this will be either "Local" or "ERA5_Land" | No |
| TEM_DATA_SOURCE | - | Declaration of the source of temperature data. Currently this will be either "Local" or "ERA5_Land" | No |

| | | | |
|---|---|---|---|
| BETA_COEFF | - | Store of the calculated beta coefficient (see pressure calculations) for each individual site | No |
| REFERENCE_PRESS | hPa | Reference pressure calculated using elevation | No |
| BD_ISRIC | $g/cm^3$ | [Bulk Density estimates taken from the International Soil Reference and Information Centre (SoilGrids250m https://soilgrids.org/ )](https://soilgrids.org/) | No |
| SOC_ISRIC | $g/dm^3$ | Soil Organic Carbon estimates from ISRIC | No |
| pH_H20_ISRIC | pH | pH of water estimates from ISRIC | No |
| CEC_ISRIC | mmol(c )/kg | Cation exchange capacity at ph7 from ISRIC | No |
| CFVO_ISRIC | $cm^3/dm^3$ | Coarse fragments from ISRIC | No |
| NITROGEN_ISRIC | cg/kg | Nitrogen in soil from ISRIC | No |
| SAND_ISRIC | g/kg | Sand in soil from ISRIC | No |
| SILT_ISRIC | g/kg | Silt in soil from ISRIC | No |
| CLAY_ISRIC | g/kg | Clay in soil from ISRIC | No |
| *_ISRIC_UC | varied | The uncertainty bounds of each of the ISRIC variables, in absolute terms. | No |
| TEXTURE | - | Soil texture identified from Sand/Silt/Clay percentages using the USDA soil texture triangle | No |
| WRB_ISRIC | - | World Reference Base (2006) soil class from ISRIC. Provided as a table of probable classes - this is the most probable class. | No |
| LAND_COVER | - | Land Cover type taken from Copernicus data set. | No |


**Table A4. crspy final output table from a given CRNS site. Note that there may be additional columns when run as different networks may have additional variables.**

| COLUMN NAME | UNITS | DESCRIPTION |
| --- | --- | --- |
| DT | datetime | Datetime of the observation. Format: "yyyy-mm-dd hh:mm:ss" |
| MOD | counts per hour | Moderated neutron count - |
| UNMOD | counts per hour | Unmoderated neutron count |
| PRESS | hPa | Atmospheric pressure recorded by the sensors at the site |
| TEMP | Celsius | Atmospheric temperature - if sensors are missing ERA5-Land data is used |
| I_TEM | Celsius | Internal Temperature of the sensor box |
| I_RH | % | Relative humidity inside the sensor box |
| E_TEM | Celsius | External (atmospheric) temperature |
| E_RH | % | External (atmospheric) relative humidity |
| RAIN | mm | Rainfall recorded at the site. If local data is unavailable, then ERA5-Land data will be used in its place |
| BATT | Voltage | Voltage of the battery |
| fbar | - | The pressure correction factor |
| DEWPOINT_TEMP | Celsius | Dewpoint temperature - from ERA5-Land data |
| SWE | mm | Snow water equivalent - from ERA5-Land data |
| ERA5L_PRESS | hPa | Atmospheric pressure - from ERA5-Land data |
| VP | hPa | Vapour Pressure - calculated |
| NMDB_COUNT | counts per hour | Neutron count rate from neutron monitoring database - usually Jungfraujoch |
| pv | kg/m$^3$ | Absolute humidity - calculated |
| fawv | - | The atmospheric water vapour correction factor |
| finten | - | The incoming cosmic-ray intensity correction factor |
| fagb | - | The above ground biomass correction factor |
| FLAG | - | The flag assigned to data in error (see Section 2.3 for definitions) |

| | | |
|---|---|---|
| MOD_CORR | counts per hour | The corrected neutron count rate after the correction factors have been applied |
| MOD_ERR | counts per hour | The statistical error of the neutron count rate |
| SM | volumetric soil moisture $cm^3$ $cm^{-3}$ | Estimated soil moisture |
| SM_PLUS_ERR | volumetric soil moisture $cm^3$ $cm^{-3}$ | Estimated soil moisture error above the estimated value - this is calculated by subtracting the MOD_ERR value (due to the inverse relationship) to the MOD_CORR value and calculating what the SM would then be |
| SM_MINUS_ERR | volumetric soil moisture $cm^3$ $cm^{-3}$ | Estimated soil moisture error below the estimated value - this is calculated by adding the MOD_ERR value (due to the inverse relationship) to the MOD_CORR value and calculating what the SM would then be |
| SM_12h | volumetric soil moisture $cm^3$ $cm^{-3}$ | The SM value with a 12-hour rolling average applied to it. Minimum number of values to calculate the 12-hour average is 6 hours of data within the 12-hour window |
| D86avg | cm | The depth of the measurement - taken as the depth from which 86% of neutrons are estimated to be sourced from (Schron et al., 2017) |
| D86avg_12h | cm | The D86 value with a 12-hour rolling average applied to it. Minimum number of values to calculate the 12-hour average is 6 hours of data within the 12-hour window |



# Appendix B: Examples of standard outputs of crspy

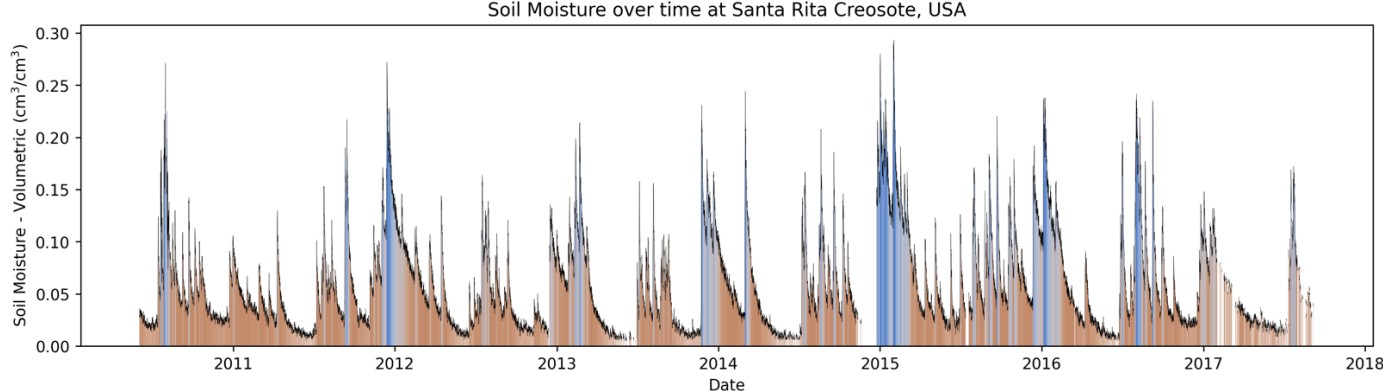


**Figure B1: Charts are output automatically that takes the fully corrected SM data and plots it over the entire time series. Optional yearly plots are also possible. The colouring is used to visually see the difference between wet (dark blue) and dry (dark brown) periods (code is found in graphical_functions.py under the colourts() function)**


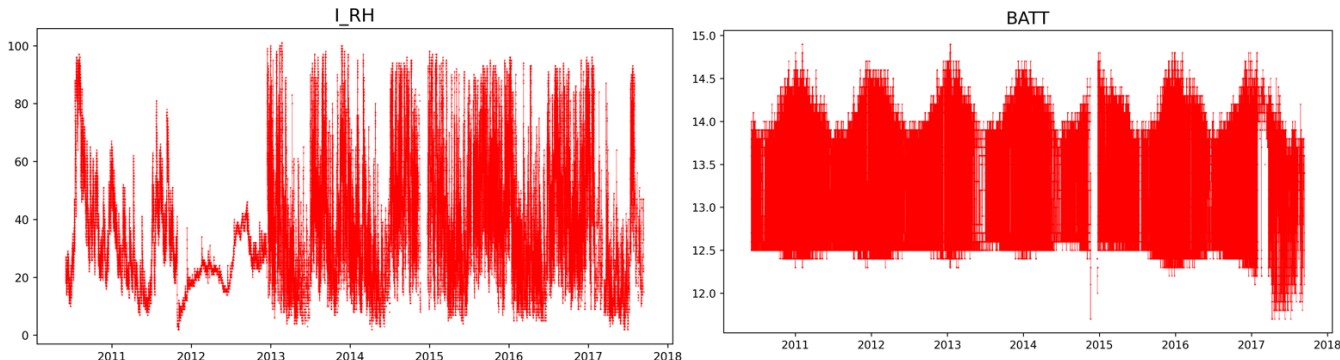

**Figure B2: Diagnostic plots are generated that creates time series of the data columns. Here two are presented (titles match variables from table A4), I_RH is the internal relative humidity and BATT is the battery voltage. These allow a user to quickly visually understand possible periods where more investigation is necessary. For example, above the**
**BATT variables begins to fall around 2017 which demonstrates an issue with the battery.**

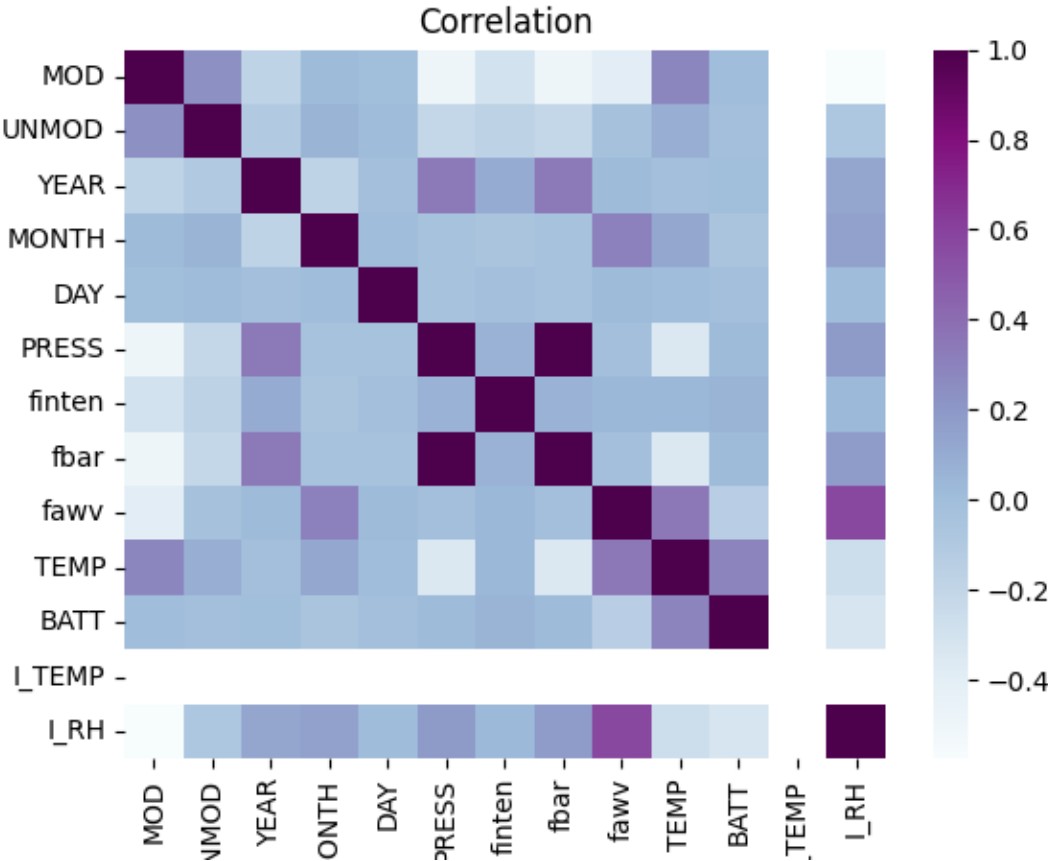

**Figure B3: A correlation heatmap is generated during quality analysis. We would expect correlation on certain variables (such as fbar and PRESS), but other correlations may point towards issues with the sensor that require investigation.**

## 7 Data availability statement

Raw CRNS data, including calibration data, is publicly available from several sources, including the US COSMOS network (http://cosmos.hwr.arizona.edu/), the Australian CosmOz (https://cosmoz.csiro.au/) and UK-COSMOS network (https://doi.org/10.5285/b5c190e4-e35d-40ea-8fbe-598da03a1185). We acknowledge the NMDB database (https://www.nmdb.eu), founded under the European Union's FP7 programme (contract no. 213007) for providing neutron count data. ESA CCI data including above ground biomass data and land cover data are available from (http://cci.esa.int/data last accessed 01/10/2021). The soil grids data are accessible online from https://soilgrids.org/. The ERA5-Land data are provided by ECMWF and are available at https://doi.org/10.24381/cds.e2161bac. The flux tower data (ARM1) is available from https://doi.org/10.17190/AMF/1246027

## 8 Code availability

The code discussed in this paper can be found at https://doi.org/10.5281/zenodo.5543669. The GitHub repository, where future updates will be uploaded can be found at https://github.com/danpower101/crspy. The GitHub repository also includes a wiki page which goes into greater detail on how to run the package. We have also generated an example walkthrough repository including example data that users can try available at: https://doi.org/10.5281/zenodo.5543652.

## 9 Author contributions

All authors were involved in the conceptualisation of this work. Data curation was undertaken by DD, SD, and DP. Investigation undertaken by RR, MRR, and DP. Software was written by DP. Manuscript drafted primarily by DP and subsequently written together with RR and MRR.

## 10 Acknowledgments

This work was funded by the Engineering and Physical Sciences Research Council (EPSRC) Water Informatics: Science and Engineering Centre for Doctoral Training (WISE-CDT) (grant number EP/L016214/1) with additional support from the Brazilian Experimental datasets for MUlti-Scale interactions in the critical zone under Extreme Drought (BEMUSED) (grant number NE/R004897/1) and the MOSAIC Digital Environment Feasibility Study (grant number NE/T005645/1), both projects both funded by Natural Environment Research Council (NERC); as well as with the support from the International Atomic Energy Agency of the United Nations (IAEA/UN) (project CRP D12014). Funding for the AmeriFlux data portal was provided by the U.S. Department of Energy Office of Science. We would also like to acknowledge GMD editor David Lawrence and three anonymous reviewers for their positive comments and very useful suggestions to improve this paper.

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
