# Peer review of "Cosmic-Ray neutron Sensor PYthon tool (crspy 1.2.1): An opensource tool for the processing of cosmic-ray neutron and soil moisture data"

_Geoscientific Model Development, 2021_

## Referee Comment (RC1)

Review of: Cosmic-Ray neutron Sensor PYthon tool (crspy): An open-source tool for the processing of cosmic-ray neutron and soil moisture data

Authors: Power et al 2021

GENERAL COMMENTS

I found this paper to be very interesting and generally well written. I think the subject matter will be increasing importance to the global CRNS community as we strive to make our datasets available and useful to the global research community. Your approach should prompt networks (existing and evolving) to think about the types of data and metadata that they would need to contribute to help harmonization.

I really like the ideas and concept presented. Those who run CRNS networks will acknowledge that their processing is not up to date but will also point out that changing a database can be a big undertaking. I think it may be worth mentioning this as a discussion point and stating that a central approach to processing might be quite valuable. It's not hard to imagine a system where networks might collect the raw data and metadata but then use crspy (or similar) as an internal processing tool to deliver the final product through their website. This is taking the product further than the intention of this paper, but it will get readers thinking. If new corrections or procedures are developed, then all that changes is a new crspy procedure calculation.

The approach of bringing in other data sets like ERA-5 and soils data to help with corrections is a great approach. Many countries are improving the spatial and temporal data sets of climate and soil properties so being able to choose a specific dataset set could be a further development for the future – again it would be good to have some brief discussion around this. This type of thing would not necessarily be for the authors to handle but a network may choose to contribute code to achieve this. This does get away from the harmonization idea, but it does open up the options further.

Related to the previous point, In terms of processing, I think you could propose two potential paths, 1) crns researcher level - the user steps through and chose the correction/datasets to apply at each step which keeps it flexible, 2) CRNS output user -global best practice which can be used for global or standardised comparisons

You say crspy can process using the most current methods – I think the issue may become keeping track of what is the "most current" method. If there is a globally accepted best approach that is going to require some discussion and agreement between network representatives. The CRNS community stands to benefit from this type of approach but some consensus on when and how to implement 'best practice' will be needed. The continued update of crspy will also need to be supported. This is an important point to make.

It would be good to see some discussion on what the future potential/ direction might be. Crspy requires a lot of user setup, package installation and folder structuring that might be beyond data users (i.e. not CRNS researchers). I had a quick go at getting crspy to run in Python a couple of months ago and ran into a couple of hurdles that stopped me proceeding through lack of time. I have limited exposure to Python having trained in R so I think most of the issues come back to my experience. That being said there could be room for some discussion around the potential for lowering the bar to entry by utilising a webpage interface. I have seen some nice Python Dash or R Shiny  applications which really make these types of things a breeze.

In summary a nice piece of work. I have some specific comments below.

SPECIFIC COMMENTS

L46 – use either "Cosmic-Ray Neutron Sensors (CRNS) are a relatively new…" or "Cosmic-Ray Neutron Sensing (CRNS) is a relatively new…"

L49/50 – this sentence makes no sense

EQ1 - May be worth noting that modification to this key equation have been very recently published (Kohli et al 2021 https://doi.org/10.3389/frwa.2020.544847 ) but not widely used. This is also highlights how your package could be useful as knowledge improves and processing evolves.

L59 – misspelling "corrections"

L71 – delete "in" before Australia

L78 – "As a consequence…"

L80 – "across" or "between" rather than "among"?

L89 – change ref style to name out side bracket - Dirmeyer et al. (2016)

L99 – This bit is a bit clunky. How about something like "Schrön et al., (2017) provided an improved approach to CRNS calibration demonstrating that their revised approach improves accuracy of soil moisture estimates. Using UK sites as an example, Schrön et al., (2017) found that …"

L102 – "…however this revised approach has not yet been deployed/applied across networks."

L121-122 – sense need rewording to make sense

L126 – this doesn't actually apply here in the text but when I looked at Table A1 to see the labelling the time zone was not specified. IF this is to be global then UTC probably needs to be specified. Or at very least have a metadata entry for time zone

Eq2 - This already highlights an issue of needed an agreed best practice. The equation noted has be widely used but I can think of alternatives already in the CRNS literature. Eq. Franz et al. 2016 Eq 2

Franz TE et al. (2016) Using Cosmic-Ray Neutron Probes to Monitor Landscape Scale Soil Water Content in Mixed Land Use Agricultural Systems. Applied and Environmental Soil Science 2016:11 doi:10.1155/2016/4323742

L169 – some success with clay content and lattice water in Australia (McJAnnet et al 2017) and limited in US (Avery et al 2016). With a global data base (which has been discussed) this could evolve – again requires cooperation between networks)

McJannet D, Hawdon A, Baker B, Renzullo L, Searle R (2017) Multiscale soil moisture estimates using static and roving cosmic-ray soil moisture sensors. Hydrol Earth Syst Sci 21:6049-6067 doi:10.5194/hess-21-6049-2017

Avery WA et al. (2016) Incorporation of globally available datasets into the roving cosmic-ray neutron probe method for estimating field-scale soil water content. Hydrol Earth Syst Sci 20:3859-3872 doi:10.5194/hess-20-3859-2016

L198 "...potential of a..." ?

L282 word missing between static and estimated?

L310 considers not considered

L353 – are these country codes from some international standard list e.g. ISO 3166 – would be useful. Is so please say which list or ISO

L472 – "...due to the fact..."

L480 – I think it should be "affect conclusions".

Fig 4 – I assume this box plots are counts of sites? This could be made clearer on the plot or the caption

L546 – last sentence is clunky and should be reworded. I assume you mean something along the lines of "Crspy has been developed to show the potential for easily and efficiently processing CRNS data in a consistent manner. The aim is to promote the usefulness of free and open access data and engage the CR"NS and research communities in the continued improvement of this product in the coming years

---

## Author Comment (AC1)

GENERAL COMMENTS

I found this paper to be very interesting and generally well written. I think the subject matter will be increasing importance to the global CRNS community as we strive to make our datasets available and useful to the global research community. Your approach should prompt networks (existing and evolving) to think about the types of data and metadata that they would need to contribute to help harmonization.

We thank the reviewer for their very positive comments. The crspy tool has been developed to help further disseminate the Cosmic-ray Neutron Sensor (CRNS) technology by providing valuable datasets for the global research community.

I really like the ideas and concept presented. Those who run CRNS networks will acknowledge that their processing is not up to date but will also point out that changing a database can be a big undertaking. I think it may be worth mentioning this as a discussion point and stating that a central approach to processing might be quite valuable. It's not hard to imagine a system where networks might collect the raw data and metadata but then use crspy (or similar) as an internal processing tool to deliver the final product through their website. This is taking the product further than the intention of this paper, but it will get readers thinking. If new corrections or procedures are developed, then all that changes is a new crspy procedure calculation.

We agree with the point raised by the reviewer that it is not so easy to change already established databases. Hence, we have developed crspy as a "post-processing" tool (i.e., raw data collection is still carried out by site PIs and networks individually). We presented the initial application utilising freely available raw data already provided by many CRNS networks.

The crspy tool can in theory provide the initial steps in developing a more centralized processing point, as described by the reviewer. However, such future version is somewhat out of the scope of this manuscript as it would require a stronger engagement by the CRNS community. This version, however, takes the first steps in providing an open-source tool in the pursuit of harmonization of CRNS datasets.

The approach of bringing in other data sets like ERA-5 and soils data to help with corrections is a great approach. Many countries are improving the spatial and temporal data sets of climate and soil properties so being able to choose a specific dataset set could be a further development for the future–again it would be good to have some brief discussion around this. This type of thing would not necessarily be for the authors to handle but a network may choose to contribute code to achieve this. This does get away from the harmonization idea, but it does open up the options further.

Related to the previous point, in terms of processing, I think you could propose two potential paths, 1) crns researcher level -the user steps through and chose the correction/datasets to apply at each step which keeps it flexible, 2) CRNS output user -global best practice which can be used for global or standardised comparisons

This is an interesting point. The use of ERA-5 land was motivated by two reasons: (1) to establish a uniform base product that can be applied on any CRNS site, and (2) to allow us to process global CRNS sites regardless of their national network affiliation. Developing an

open source allows users to implement their own routines to include alternative data sets as needed. We will discuss this opportunity in the update manuscript.

The two proposed paths are attractive, and we would suggest that currently crspy's focus is on being a research level tool that allows data harmonization to begin comparing sites as a global network. We agree that having a global best practice option would be of great benefit, however this will require input and discussion from numerous groups and researchers. If a common best practice for CRNS data processing is decided by the community in the future, crspy can include this as it's default process.

You say crspy can process using the most current methods –I think the issue may become keeping track of what is the "most current" method. If there is a globally accepted best approach that is going to require some discussion and agreement between network representatives. The CRNS community stands to benefit from this type of approach but some consensus on when and how to implement 'best practice 'will be needed. The continued update of crspy will also need to be supported. This is an important point to make.

As above, we agree with the points made here and hope that future discussion and collaboration within the CRNS community can lead to an established 'best practice'. Our aim is to continue updating crspy in the future to keep pace with advancements. In that sense, future developments of crspy can be interpreted, by analogy, as following similar steps when hydrological or land surface models incorporate new parameterizations due to new knowledge available from the community. crspy's list of functions and subroutines have been designed to account for future updates which can be replaced or updated easily.

It would be good to see some discussion on what the future potential/ direction might be. Crspy requires a lot of user setup, package installation and folder structuring that might be beyond data users (i.e. not CRNS researchers). I had a quick go at getting crspy to run in Python a couple of months ago and ran into a couple of hurdles that stopped me proceeding through lack of time. I have limited exposure to Python having trained in R so I think most of the issues come back to my experience. That being said there could be room for some discussion around the potential for lowering the bar to entry by utilising a webpage interface. I have seen some nice Python Dash or R Shiny applications which really make these types of things a breeze.

We first thank the reviewer for trying crspy and already giving us some user-specific feedback. This is very important for us to provide a clean and easy-to-use tool to the community. We welcome the reviewer's suggestion of adding more discussion on the future direction with crspy. Indeed, the set-up process may not be fully suitable for entry level users. We aim to collect feedback from users in order to understand how we can improve crspy in future versions, although currently we would consider it a research level tool. One particular issue we have encountered when developing crspy is the wide range of formatting styles of data from the different networks (i.e., column orders, column titles etc). This ultimately means additional work for the user to prepare the data environment initially. Standardisation of data structure could be an option in the future, such as is currently done in the Fluxnet community but would require the national networks to define and adhere to such standard.

In summary a nice piece of work.

We once again thank the reviewer for their very positive feedback. We will also deal with more specific comments raised by the reviewer (not included in this response) in the revised version of the manuscript.

---

## Author Comment (AC2)

Response to Reviewer 2

This paper describes an open-source python tool called crspy that is designed to facilitate the processing of raw CRNS data into soil moisture estimates in an easy and harmonized way. Although the tool I think is useful, more explanations about the data inputs and data fusion methods and the applications and comparisons with other existing models are needed. Please see my comments below:

We thank the reviewer for taking the time to review our manuscript. The comments and suggestions provided will help us to improve it in the next iteration. Our replies to your specific comments are below.

The spatial mismatch between the ERA-5 land and the CRNS datasets is quite large (0.6 km vs 9 km). Before applying the ERA-5 data directly into your modelling, has the data been evaluated against the in-situ data first? For users/readers, it's useful to know this information.

We have not directly evaluated the ERA5-Land data with in-situ data in this study. We will make this clear in the updated manuscript.

Can you explain more about how the ERA-5 data are used for filling in the data gap? Which data fusion method is used in your tool? How did you tackle the spatial mismatching issue?

The main reason behind introducing this routine was due to the fact many of the earliest CRNS sensors do not include external sensors to measure standard meteorological variables such as air temperature and humidity. We now know that such additional measurements are essential to account for the influence of atmospheric water vapour dynamics on the neutron count rate (e.g., US COSMOS). We have previously found that neglecting to apply the water vapor correction can lead to soil moisture errors on the order of 30%-50% for sites where high atmospheric water vapor seasonality is observed (Rosolem et al. 2013). We selected ERA5-Land as a replacement for those external data in a similar way to, for example, synthesis datasets from the Fluxnet network uses the ERA-Interim dataset. Notice however that in our case, the sites will not necessarily have local data to be used for downscaling. We will discuss this further in the revised version of the manuscript highlighting the advantages and disadvantages.

Rosolem, R., Shuttleworth, W. J., Zreda, M., Franz, T. E., Zeng, X., & Kurc, S. A. (2013). The effect of atmospheric water vapor on neutron count in the cosmic-ray soil moisture observing system. Journal of Hydrometeorology, 14(5), 1659–1671. https://doi.org/10.1175/JHM-D-12-0120.1

It is stressed by the authors that the intention of the work is not to identify which method is better or worse than the other. This is a bit confusing as if we (users) don't know the comparative performance, how can we be confident in choosing your model. They can choose a more accurate model which I think is as important as the harmonized step.

We consider it outside the scope of this paper to rank the different networks on performance with regards to the chosen processing steps. We believe that all networks have significant contribution to the wider community in providing soil moisture data from this recent technology. As discussed in our reply to Reviewer#1, a community driven best practice method is probably the best outcome for a global network of such sensors, and crspy can facilitate the steps towards achieving such goals. We will discuss the future direction of crspy in the updated manuscript, which includes maintaining it with the most current methods based on our developing understanding of the CRNS method.

Pg 10-11. "The data required for the calibration step includes the date of …. volumetric soil moisture of the sample." Where are the sensor's calibration data from? Are these the information already available with all the existing Cosmic-Ray sensors around the world?

We thank the reviewer for raising this point which is likely due to lack of clarity. The calibration data is providing directly by the  user typically from many soil samples taken at the site. Notice that some of the networks provide this freely and we will make this clearer in the updated manuscript.

We once again thank the reviewer for providing us with valuable feedback.

---

## Author Comment (AC3)

Response to reviewer #3

The manuscript presents a new processing tool for stationary cosmic-ray neutron sensors based on a python package on GitHub. The tool is capable of reading in CRNS time series data and soil sampling data in a given data format, of filtering and correcting the data, and of generating an output that includes data products like soil moisture, uncertainty, and penetration depth. The tool is also capable of consulting external data, like ERA5, to support the gap filling and meta data description. The author's vision is that processing steps should be harmonized across all CRNS networks and that the community of users and researchers will use and maintain this code to generate their data products.

I do fully support this vision and I agree that it is about time to provide researchers and users a tool to more efficiently and consistently work with CRNS data. Crspy is one of the first open-source tools that offers a timely and substantial contribution towards this goal and hence it is worth to be published in GMD.

We thank the reviewer for taking the time to review our manuscript and for their positive feedback. Their comments and suggestions provided will help us to improve our manuscript in the next iteration. We are also pleased that they also share in our vision to provide open tools for the CRNS community.

**General concerns**

1. The manuscript is consice and well written, but my impression is that it falls short of more elaborate explanations regarding (1) the technical details how Crspy works, and (2) regarding user guidance.

2. From a GMD paper I would expect that every single equation and processing step is explicitly described and mathematically clear. This will allow users to fully understand what the model does without looking at the code. Hence, I strongly suggest that all these parts – e.g. about the air pressure correction, the soil sample calibration, or the temporal aggregation, to name just a few – should be much more elaborated. Essentially, it would require not much more than typesetting the procedures used in the code. But from my understanding this is standard for articles on new tools and models.

   In producing this manuscript we wanted to ensure that it is informative and readable and so we have described the key equations that will impact the outputs. Although we agree it is important for users to understand all aspects of the way a model works, if we were to typeset every single equation, we feel it would make the paper unreasonably large. In addition, many implementations of routines are from established literature, and we feel to be sufficient to point the general GMD readership to the appropriate papers without reproducing the same set of equations here, hence focusing on the

integrating aspect of crspy as a data processing tool. We agree however that we should ensure it is clear on any decision points made, such as aggregation methods, to give users a clear understanding of how the data is processed. We will address this in the updated manuscript.

3. An important detail which directly follows from the previous comment is that it was not clear to me from reading the manuscript how aggregation and/or smoothing of the data is performed. Do you aggregate neutrons before conversion to soil moisture, or do you aggregate the final soil moisture product? Is the aggregated data indexed at the start, middle, or end of the aggregated period? These details sound picky in the first place, but they are of major importance since they can have substantial effect on the final product (due to the non-linearity of theta(N)) and on the comparability to other processing tools.

   We thank the reviewer for raising this comment which we believe I might have been due to lack of clarity in our manuscript. Currently we aggregate the soil moisture estimates after processing (i.e. each hourly reading is processed to give a soil moisture estimate, any aggregation is then undertaken on the soil moisture values with the indexing occurring on the end of the aggregated period). For the smoothing we have employed a moving 12-hour average (with a constraint that there is a minimum of 6 hours of data in the window). This 12-hour window is well established in the cosmic-ray neutron sensing community (e.g., COSMOS) and also confirmed by our preliminary analysis at our UK sites (possibly one of the most challenging ones due to soil wetness, proximity to sea level, and humid atmosphere) showing 12-hour averaging to satisfactory (data not shown). We will be clearer about these points in the updated manuscript.

4. I would suggest that the manuscript elaborates a little bit more on the details of how crspy works internally and how it should be maintained by the community. Not because the needs of expert programmers should be addressed, but rather to facilitate community-driven updates of the code. Since the CRNS research changes their methods often, it would be a key feature of crspy to be adaptable by the community. So please provide a key section on (1) guiding researchers how the code could be changed, e.g., if a new correction function needs to be included, and (2) guiding users what to do if they want to use the new correction (update the script, change meta data, etc.). Add also dicussion on how can the community make sure that scientists regularly update their code? How can users of the data verify the the processing scheme of a data is up to date?

   This is a great idea and something we will address in the updated manuscript. We have already included on the github page an example

workflow that is intended to help users understand how to run crspy (from installation through to processing). We have also spent time ensuring that the code is thoroughly commented. Including a key section as described above would help to bridge the gap for those who wish to use and alter crspy in the future.

5. My impression is that the authors undersell Crspy in this short manuscript. It looks like crspy has a lot of useful features and products, which are only marginally mentioned in the text and figures. I would suggest to more prominantly illustrate potential data products of crspy, e.g., a soil moisture time series including their error band, the footprint depth, examples of flagged data in certain periods, or diagnostic output. Moreover, it is very promising to see that the metadata can be used to do meta analysis on the data, but you only show examples using land use or meteorological data. From my perspective, the meta data analysis would be even more valuable for the CRNS community when looking at site-specific paramaters, soil properties, and their correlation to N0, GV, or biomass, for instance. I'd recommend to also provide such an example (similar to what was used in Shuttleworth et al. 2013 to correlate COSMIC parameters with soil bulk density), as this would push the community research a lot forward.

We thank the reviewer for this very positive comment. When writing the manuscript, we have tried to balance demonstrating all the features whilst ensuring it remains concise and readable. We will improve on this in the next iteration through additional figures that highlight some of the features mentioned above.

Currently, we have initially addressed the main comments made by the reviewer. We will respond to all technical and more specific comments raised by the reviewer should a revised version of the manuscript be invited by the GMD Editor.

We once again thank the reviewer for the very positive feedback and their comments and questions.

---

## Author Response (AR1)

We thank the reviewers for their comments, suggestions and valuable feedback. This has helped us to improve the manuscript. The reviewers comments are shown in black and the replies are shown in red.

**Reply to Reviewer 1**

Review of: Cosmic-Ray neutron Sensor PYthon tool (crspy): An open-source tool for the processing of cosmic-ray neutron and soil moisture data

Authors: Power et al 2021

GENERAL COMMENTS

I found this paper to be very interesting and generally well written. I think the subject matter will be increasing importance to the global CRNS community as we strive to make our datasets available and useful to the global research community. Your approach should prompt networks (existing and evolving) to think about the types of data and metadata that they would need to contribute to help harmonization.

I really like the ideas and concept presented. Those who run CRNS networks will acknowledge that their processing is not up to date but will also point out that changing a database can be a big undertaking. I think it may be worth mentioning this as a discussion point and stating that a central approach to processing might be quite valuable. It's not hard to imagine a system where networks might collect the raw data and metadata but then use crspy (or similar) as an internal processing tool to deliver the final product through their website. This is taking the product further than the intention of this paper, but it will get readers thinking. If new corrections or procedures are developed, then all that changes is a new crspy procedure calculation.

We thank the reviewer for their very positive comments about crspy and helpful feedback. We have updated the manuscript (L85) to discuss the issues that are encountered in updating a database compared to updating datasets.

The approach of bringing in other data sets like ERA-5 and soils data to help with corrections is a great approach. Many countries are improving the spatial and temporal data sets of climate and soil properties so being able to choose a specific dataset set could be a further development for the future – again it would be good to have some brief discussion around this. This type of thing would not necessarily be for the authors to handle but a network may choose to contribute code to achieve this. This does get away from the harmonization idea, but it does open up the options further.

This is an important point in which we agree, hence we have developed crspy as an open package for the community. For additional information, we have included supplementary data that outlines the structure of crspy and gives an example of how to update the code. This is to give a better understanding of how one might change the code for their own use, for example to integrate alternative products as suggested above.

Related to the previous point, In terms of processing, I think you could propose two potential paths, 1) crns researcher level - the user steps through and chose the correction/datasets to apply at each

step which keeps it flexible, 2) CRNS output user -global best practice which can be used for global or standardised comparisons

We thank the reviewer for this suggestion. We have added this discussion in future directions.

You say crspy can process using the most current methods – I think the issue may become keeping track of what is the "most current" method. If there is a globally accepted best approach that is going to require some discussion and agreement between network representatives. The CRNS community stands to benefit from this type of approach but some consensus on when and how to implement 'best practice' will be needed. The continued update of crspy will also need to be supported. This is an important point to make.

We agree with the reviewers comments here that there are difficulties in establishing what the "most current" method is. We have updated with the future directions sections where we also emphasise that a standard method agreed within the whole community will be important going forward.

It would be good to see some discussion on what the future potential/ direction might be. Crspy requires a lot of user setup, package installation and folder structuring that might be beyond data users (i.e. not CRNS researchers). I had a quick go at getting crspy to run in Python a couple of months ago and ran into a couple of hurdles that stopped me proceeding through lack of time. I have limited exposure to Python having trained in R so I think most of the issues come back to my experience. That being said there could be room for some discussion around the potential for lowering the bar to entry by utilising a webpage interface. I have seen some nice Python Dash or R Shiny applications which really make these types of things a breeze.

We thank the reviewer for these suggestions. The issue of making crspy more accessible and user friendly is something that we are continuing to improve upon and has also been mentioned by Reviewer 3. We have made improvements to the way that crspy is installed (such as being able to install it through PyPi (https://pypi.org) as well as changing the way the configuration file is imported), that we hope can begin to address some of the mentioned issues. This is also mentioned in the newly updated walkthrough document uploaded as supplementary data.

We have also begun development of a version of crspy with Docker (https://www.docker.com) which should alleviate some issues with system dependencies (although this would require a knowledge of Docker to run). Additional applications such as Python Dash would be welcome additions and something we could consider for future versions of crspy. It is important to also recognize that future developments of crspy, both in terms of its science as well as its structure, will be undertaken following direct feedback from the community and early adopters.

In summary a nice piece of work. I have some specific comments below.

We once again thank the reviewer for their very positive feedback and very helpful suggestions.

SPECIFIC COMMENTS

L46 – use either "Cosmic-Ray Neutron Sensors (CRNS) are a relatively new…" or "Cosmic-Ray Neutron Sensing (CRNS) is a relatively new…"

Fixed

L49/50 – this sentence makes no sense

Fixed

EQ1 - May be worth noting that modification to this key equation have been very recently published (Kohli et al 2021 https://doi.org/10.3389/frwa.2020.544847 ) but not widely used. This is also highlights how your package could be useful as knowledge improves and processing evolves.

Discussed in future directions

L59 – misspelling "corrections"

Fixed

L71 – delete "in" before Australia

Done

L78 – "As a consequence…"

Done

L80 – "across" or "between" rather than "among"?

Done

L89 – change ref style to name out side bracket - Dirmeyer et al. (2016)

Fixed

L99 – This bit is a bit clunky. How about something like "Schrön et al., (2017) provided an improved approach to CRNS calibration demonstrating that their revised approach improves accuracy of soil moisture estimates. Using UK sites as an example, Schrön et al., (2017) found that …"

Re-worded sentence

L102 – "…however this revised approach has not yet been deployed/applied across networks."

Adjusted sentence along with additional references

L121-122 – sense need rewording to make sense

Re-worded for clarity

L126 – this doesn't actually apply here in the text but when I looked at Table A1 to see the labelling the time zone was not specified. IF this is to be global then UTC probably needs to be specified. Or at very least have a metadata entry for time zone

This is a good point and one mentioned by Reviewer 3 too. We will ensure it is clear that UTC time should be the standard in table A1. Time zone is included in metadata should external data products require adjustment to UTC.

Eq2 - This already highlights an issue of needed an agreed best practice. The equation noted has be widely used but I can think of alternatives already in the CRNS literature. Eq. Franz et al. 2016 Eq 2

Franz TE et al. (2016) Using Cosmic-Ray Neutron Probes to Monitor Landscape Scale Soil Water Content in Mixed Land Use Agricultural Systems. Applied and Environmental Soil Science 2016:11 doi:10.1155/2016/4323742

We agree with the reviewer here that a best practice will be important going forward.

L169 – some success with clay content and lattice water in Australia (McJAnnet et al 2017) and limited in US (Avery et al 2016). With a global data base (which has been discussed) this could evolve – again requires cooperation between networks)

McJannet D, Hawdon A, Baker B, Renzullo L, Searle R (2017) Multiscale soil moisture estimates using static and roving cosmic-ray soil moisture sensors. Hydrol Earth Syst Sci 21:6049-6067 doi:10.5194/hess-21-6049-2017 Avery WA et al. (2016) Incorporation of globally available datasets into the roving cosmic-ray neutron probe method for estimating field-scale soil water content. Hydrol Earth Syst Sci 20:3859-3872 doi:10.5194/hess-203859-2016

We thank the reviewer for pointing this out. We have updated the manuscript to reflect that there are techniques available to estimate lattice water content and included the provided references.

L198 "…potential of a…" ?

Fixed

L282 word missing between static and estimated?

Fixed

L310 considers not considered

Fixed

L353 – are these country codes from some international standard list e.g. ISO 3166 – would be useful. Is so please say which list or ISO

Currently there is no defined standard for crspy. For the code to function any string will be readable as long as both the name and metadata are the same. In a push towards standardisation however, this could be something worth considering, but it is outside the scope of this paper.

L472 – "…due to the fact…"

Fixed

L480 – I think it should be "affect conclusions".

Done

Fig 4 – I assume this box plots are counts of sites? This could be made clearer on the plot or the caption

Updated the caption to be clearer.

L546 – last sentence is clunky and should be reworded. I assume you mean something along the lines of "Crspy has been developed to show the potential for easily and efficiently processing CRNS data in a consistent manner. The aim is to promote the usefulness of free and open access data and engage the CRNS and research communities in the continued improvement of this product in the coming years

Agree that re-wording will be clearer on the aims.

Original:

crspy has been developed to open up the debate and use of free and public open CRNS data, and we invite the general community to engage with us to improve this platform in future years.

New:

crspy has been developed to show the potential for easily and efficiently processing CRNS data in a harmonized way. The aim is to promote the usefulness of free and open access data and engage the CRNS and research communities in the continued improvement of this product in the coming years.

This paper describes an open-source python tool called crspy that is designed to facilitate the processing of raw CRNS data into soil moisture estimates in an easy and harmonized way. Although the tool I think is useful, more explanations about the data inputs and data fusion methods and the applications and comparisons with other existing models are needed. Please see my comments below:

The spatial mismatch between the ERA-5 land and the CRNS datasets is quite large (0.6 km vs 9 km). Before applying the ERA-5 data directly into your modelling, has the data been evaluated against the in-situ data first? For users/readers, it's useful to know this information.

Can you explain more about how the ERA-5 data are used for filling in the data gap? Which data fusion method is used in your tool? How did you tackle the spatial mismatching issue?

Currently ERA5-Land data has not been evaluated against in-situ data for this study. The main reason being that where we require the use of ERA5-Land (i.e., US COSMOS network) we often do not have in-situ data available for comparison. Unlike other networks that were established afterwards, the US COSMOS stations did not include co-located standard meteorological measurements with the cosmic-ray neutron sensors, apart from selected Ameriflux sites. However, the merging between US COSMOS and Ameriflux is currently beyond the scope of this manuscript and not pursued in version 1.0 of crspy for simplicity. Currently ERA5-Land data is used to replace missing sensors. For example, some earlier sites do not have external temperature sensors, as we now need this to correct for atmospheric water vapour, we use ERA5-Land data in place of the sensor.

We agree with the reviewer that some kind of bias correction could improve the model when using reanalysis data, however currently this is outside the scope of crspy. We believe that using reanalysis data is better than not correcting for influences on the signal at all. We will discuss this in the future direction of crspy as this is something that could be worth exploring in the future.

To demonstrate the benefit of using ERA5-Land even when no bias correction is possible, we have processed one of the USA sites (ARM-1 from the COSMOS network) in three ways. This site is co-located with a flux tower which has in-situ measurements for relative humidity and atmospheric temperature. We manually replaced the missing columns in the CRNS data with this in-situ data and processed it through crspy. This could be considered the best method using "in-situ" data.

We then processed the same site using; 1) un-bias corrected ERA5-Land data to correct for atmospheric water vapour (using Temperature and Dewpoint Temperature) or 2) without a correction for atmospheric water vapour at all. The figures below shows the difference of SM_12h (the estimated soil moisture filtered with a 12-hour rolling average) from the ground truth (using in-situ data) and the two alternative approaches. We can see that when there is no correction for atmospheric water vapour there is a seasonal aspect to the error, and that it is much more pronounced with a broader spread in error values. There is still some error when using method 1) however this is much lower in magnitude.

[Figure]

**Figure 1 shows the difference between the 12h rolling average of SM estimates using in-situ data as a base and in blue, using ERA5-Land data without bias correction, and in red, no correction for fawv at all. The x-axis represents time steps in hours from 2010-07-16 to 2018-10-25.**

[Figure]

**Figure 2 shows the same data as figure 1 but in a box chart format. The whiskers are set to the 5th and 95th percentiles with the dots representing the outliers. Here we can clearly see that using ERA5-Land data that is not bias corrected leads to much less error (compared to in-situ data) as opposed to not applying the correction at all.**

It is stressed by the authors that the intention of the work is not to identify which method is better or worse than the other. This is a bit confusing as if we (users) don't know the comparative performance, how can we be confident in choosing your model. They can choose a more accurate model which I think is as important as the harmonized step.

We developed crspy as a tool to easily process CRNS data and have shown the differences between networks to demonstrate why we believe a standardised methodology is important. We don't believe it is for us as authors of this tool to decide on which method is best, as currently there is literature demonstrating the benefits of each method (Hawdon et al., 2014, Evans et al., 2016). However, crspy is developed by us following what we currently believed to be the minimum standards required to process cosmic-ray neutron sensors, especially if a multi-site analysis is to be undertaken. In addition, we have outlined our opinion in the future directions section that moving forward a standard approach should be the goal of the CRNS community.

Pg 10-11. "The data required for the calibration step includes the date of …. volumetric soil moisture of the sample." Where are the sensor's calibration data from? Are these the information already available with all the existing Cosmic-Ray sensors around the world

We apologize to the reviewer for the confusion. We have added a brief explanation to emphasise that some calibration data will be available at established sites (L332) and point to resources about calibration. We have also updated the data availability statement to be clearer.

The manuscript presents a new processing tool for stationary cosmic-ray neutron sensors based on a python package on GitHub. The tool is capable of reading in CRNS time series data and soil sampling data in a given data format, of filtering and correcting the data, and of generating an output that includes data products like soil moisture, uncertainty, and penetration depth. The tool is also capable of consulting external data, like ERA5, to support the gap filling and meta data description. The author's vision is that processing steps should be harmonized across all CRNS networks and that the community of users and researchers will use and maintain this code to generate their data products.

I do fully support this vision and I agree that it is about time to provide researchers and users a tool to more efficiently and consistently work with CRNS data. Crspy is one of the first open-source tools that offers a timely and substantial contribution towards this goal and hence it is worth to be published in GMD.

We thank the reviewer for their positive comments recognizing crspy as one of the first open-source tools aimed at processing cosmic-ray neutron sensors for soil moisture estimates. We also thank the reviewer for their useful feedback.

**General concerns**

The manuscript is consice and well written, but my impression is that it falls short of more elaborate explanations regarding (1) the technical details how Crspy works, and (2) regarding user guidance.

From a GMD paper I would expect that every single equation and processing step is explicitely described and mathematically clear. This will allow users to fully understand what the model does without looking at the code. Hence, I strongly suggest that all these parts – e.g. about the air pressure correction, the soil sample calibration, or the temporal aggregation, to name just a few – should be much more elaborated.

Essentially, it would require not much more than typesetting the procedures used in the code. But from my understanding this is standard for articles on new tools and models.

We thank the reviewer for their comments on this point. We also recognize the importance of being clear about our developed software. Due to the fact that it is a framework that implements equations and processes aggregated from many different sources over years we have tried to keep the manuscript concise enough to be readable whilst being clear about what is being done. We have used the main neutron-to-soil moisture equation (Equation 1) as the reference point throughout the paper to provide the reader with all available information regarding the steps as well as pointing out to relevant literature for any intermediate steps undertaken within crspy. In addition to the literature, more information is presented in crspy's Github website for users. However, upon reading the reviewer comments we have spent some time creating supplementary document that describes the steps undertaken in the code when running the main function. This is in order to bridge the gap between the manuscript and the actual code (which we have also endeavoured to ensure is clearly commented throughout). Our aim with this supplementary document is to address the point about helping users understand the technical details of how the code works and how they may change it themselves, available at (

On the second point about user guidance, we have also included a supplementary repository that includes an IPython notebook along with example datasets that a user can work through to have a demonstration of how crspy can process data (available at https://doi.org/10.5281/zenodo.5484102)

An important detail which directly follows from the previous comment is that it was not clear to me from reading the manuscript how aggregation and/or smoothing of the data is performed. Do you aggregate neutrons before conversion to soil moisture, or do you aggregate the final soil moisture product? Is the aggregated data indexed at the start, middle, or end of the aggregated period? These details sound picky in the first place, but they are of major importance since they can have substantial effect on the final product (due to the non-linearity of theta(N)) and on the comparability to other processing tools.

The aggregation of soil moisture data is done after the conversion of hourly neutrons to hourly soil moisture estimates. That is, a rolling average is applied to the hourly soil moisture estimates and not to the neutron counts. The rolling average is set to 12-hours following the methods originally applied in the COSMOS network (Zreda et al., 2012). We have included a sentence that points to the supplementary data (Table A.4) describing these outputs.

I would suggest that the manuscript elaborates a little bit more on the details of how crspy works internally and how it should be maintained by the community. Not because the needs of expert programmers should be addressed, but rather to facilitate community-driven updates of the code. Since the CRNS research changes their methods often, it would be a key feature of crspy to be adaptable by the community. So please provide a key section on (1) guiding researchers how the code could be changed, e.g., if a new correction function needs to be included, and (2) guiding users what to do if they want to use the new correction (update the script, change meta data, etc.). Add also dicussion on how can the community make sure that scientists regularly update their code? How can users of the data verify the the processing scheme of a data is up to date?

We thank the reviewer for their suggestions. As discussed above we have spent some time creating a supplementary document that tries to explain the structure of the code in a readable way, that we hope will provide additional support for any user who wishes to alter crspy themselves. We agree that in order for crspy to be truly useful for the community, the community itself must be able to

alter and change it to accommodate their own needs. We have also discussed in the future directions section the importance of the community establishing a standard format.

My impression is that the authors undersell Crspy in this short manuscript. It looks like crspy has a lot of useful features and products, which are only marginally mentioned in the text and figures. I would suggest to more prominantly illustrate potential data products of crspy, e.g., a soil moisture time series including their error band, the footprint depth, examples of flagged data in certain periods, or diagnostic output. Moreover, it is very promising to see that the metadata can be used to do meta analysis on the data, but you only show examples using land use or meteorological data. From my perspective, the meta data analysis would be even more valuable for the CRNS community when looking at site-specific paramaters, soil properties, and their correlation to N0, GV, or biomass, for instance. I'd recommend to also provide such an example (similar to what was used in Shuttleworth et al. 2013 to correlate COSMIC parameters with soil bulk density), as this would push the community research a lot forward.

We thank the reviewer for these positive comments. We agree that metadata analysis can open up avenues of research that can further aid our understanding of the sensor. We have expanded the discussion in the metadata section to reflect this and point to some other ways this data can be used. We are also working on a paper that will explore this in greater detail in the future and wish to keep the focus of this paper on the tool itself. We aimed to demonstrate some examples to illustrate this. We have also added an Appendix section (Appendix B) that demonstrates some of the other outputs available in crspy as suggested above.

**Technical concern**

Naming conventions (L124: "it is first necessary for a user to correctly format input data following crspy's naming convention"): I think one of the biggest obstacle for users to apply the new tool would be that crspy requires a certain data format (Tables A1 and A2), while most data portals and CRNS data providers have already their data format fixed. I'd suggest to slightly adapt the crspy meta files such that the user may define the column name of their variable indivudally (e.g., temperature_column name = TEMP. Then, crspy can address the data by the given column name, independent of column position or other restrictions, which would facilitate much smoother integration into existing data workflows.

We appreciate that having the user change their data to fit the crspy format is an obstacle to easy implementation. The described process could be useful in addressing this but when using crspy to process sites for multiple networks this metafile would need to be changed often which could itself be problematic. Alternatively, a standardised system similar to FLUXNET would be beneficial with uniform column labels. This would require a community driven approach that we discuss in the future direction section. Should a community decision be taken, crspy can be easily modified to accommodate such changes and new standards.

The initial setup of the code requires a steep learning curve and a lot of prior knowledge. There might be ways to lower the bar for users, e.g., by providing a setup wizard script which assists the user in setting up their first connection to the data base and creation of a station and its meta data. by providing a first fully-working example with which the user could start right away after installing the tool. By deploying the project with Docker, a platfrom that unfolds automatically on any system without prior trouble regarding python installations.

We have addressed some of the issues raised by the reviewer concerning the initial set up by changing how crspy is installed. For example, we have uploaded crspy into the PyPi repository (https://pypi.org) so it can be installed with a single command `pip install crspy`. We also changed the need to copy a name_list.py file into the working directory. In place we redesigned the initial script to auto build a config.ini file. This removes steps in the working directory set up that should reduce potential issues.

We have also developed a version of crspy with Docker (https://www.docker.com) version of crspy which addresses some of the dependency issues users may run into (such as ensuring the HDF5 headers are available on the host computer system). Although currently this should be considered a development version which we will aim to improve upon in the future. Instructions and information on these revised methods are included on the GitHub page. We thank the reviewer for these suggestions as we believe they will improve the accessibility of crspy to users.

**Minor concerns**

- Eq. 9: Crspy cuts off soil moisture at the maximum porosity of the site. Does that mean that crspy cannot be used for periods of snow or ponded/intercepted water?

  We have updated the manuscript to explain how to remove this cut-off should a user wish to explore the impact of snow or ponded water.

- ERA5: while it is a good idea to fill data with ERA5, there could be a spatial mismatch of scales and also a bias in absolute values due to metroligical reasons. This could lead to a significant bias of neutrons particularly when gap-filling air pressure. Is there a way to first compare local data with ERA5 data, identify their constant bias, and then use it to gap-fill? At least it would be good for crspy to create diagnostic output showing local air pressure versus ERA5 air pressure (and humidity, temperature, …) in order to check the consistency from time to time.

  This is a good point that has been raised previously by reviewer 2 also. As discussed above currently the gap filling with ERA5-Land is being undertaken at sites without sensors to bias correct to. In the future methods to bias correct data with in-situ measurements could improve the accuracy of the soil moisture estimates. Currently we are taking an approach where we are aware that there are still gains to be made in terms of accuracy but are confident that using un-bias corrected reanalysis data is better than not correcting for these

influences on the signal at all (as shown in the figure above). In future iterations of crspy we agree that this is an area that can be further investigated.

- Line 258: When the option to find the nearest NM is used, why do you ignore the GVcorrection factor? It could still be that the nearest station is many GVs away. And, what if the nearest station has no data? (as it often happens on nmdb.eu), do you take the next nearest station?

  This is an interesting point and one that we think may require more research. Currently crspy is designed using current methodologies following for example, Hawdon et al., (2014) method which uses the nearest site without additional correction. For this initial development version, we have decided to maintain this option as it is.

- Filter: You seem to remove counts below 30% of N0, which is reasonable as one would not expect a stronger effect on the neutrons than pure water could have. But I do not agree on removing counts above N0. Under very dry contions, counts could exceed N0 by a few percent. (check N(theta=0, N0)>N0)

  This is an interesting point made by the reviewer. To accommodate that, we have currently changed crspy so that the cut-off is subjectively chosen to be N0*1.075, based on the below referenced N_max value.

- Filter: very often the imcoming correction does not work very well in periods of groundlevel enhancements from cosmic-rays or during coronal mass ejections. One could offer antoher option to flag also data where the change of neutron monitor data is suspicious, i.e., drops by a few percent from one day to another.

  We thank the reviewer for this comment. Currently, this is not possible in this version of crspy, but as we expect to update crspy in the future with increased understanding of external impacts this may change once there is a defined way to do this. We would certainly invite the Space Weather community to provide us feedback on the use of crspy. In fact, we are already involved in better understanding GLEs phenomena using cosmic-ray neutron sensors (Hands et al. 2021) in which we used a beta version of crspy previously.

**Specific comments**

Eq. 1: The authors may want to acknowledge the new formulation from Köhli et al (2021), which is equivalent to the function from Desilets et al. (2010) using N_max=1.075 N_0. It would not change anything in the results. But the Desilets equation has 4 parameters, which is one parameter more than necessary and thus the function is overstated and would lead to non-unique calibration results. The equivalent reformulation is only 3 parameters and would lead to unique optimization results if needed. Note that the other UTS equation proposed by Köhli et al. is too new to be addressed by this version of Crspy, but it is a good example of quickly changing CRNS methods, and demonstrates how important it is that Crspy should be flexible and adaptable by the community.

We have included a discussion on this new formulation in the future direction section of crspy.

Line 73: citations of networks are:

  Zacharias, S. et al. A Network of Terrestrial Environmental Observatories in

  Germany. Vadose Zone J **10**, 955–973 (2011)

  Bogena, H. R. TERENO: German network of terrestrial environmental observatories.

  J Large-scale Res Facil Jlsrf **2**, 52 (2015).

   Cooper, et al.: COSMOS-UK: national soil moisture and hydrometeorology data for environmental science research, Earth Syst. Sci. Data, 13, 1737–1757, (2021)

We have updated the manuscript to reflect these citations for the networks, thank you for pointing these out to us.

Line 102: Please reformulate "currently this revised approach is not applied across the networks", I think many recent studies adopted the new approach, and the COSMOSUK network seems to apply that the new weighting, too (please double check)

Reformulated sentence and added reference to take into account that this approach is being applied in more recent studies.

Parts of Fig 1 are hard to read, can you increase font size (e.g. reduce time stamps and number of rows in the boxes) and increase resolution (e.g. vector format, pdf/eps)?

We have updated the figure to increase the font size of the most important aspects that describe the steps.

Line 140–145: note that the equation has been shown to work not very well for very dry conditions (Köhli et al. 2021) and that the parameters a0..a2 are not always constant for all sites, since literature exists where these parameters have been adapted. Since crspy aims at offering a general solution, would there be a possibility to change this equation or to automatically fit these parameters?

We have added discussion in the new future direction section to discuss this revised approach. In this version of crspy we are using the well-established base

Eq 2: can you please provide a reference for this equation and the 0.556 factor?

Reference is from the Hawdon et al., (2014) paper, we have updated this sentence to make this clearer.

Line 170: if LW is not provided, why don't you estimate it from clay content using the soilgrids data base?

This point has also been raised by reviewer 1, we have updated the sentence here to point to the

fact that there are methods available for the estimation of LW.

Line 223: The reference from Desilets 2021 is only a technical document and it is not availabel under the given doi. Please check whether a better reference could be given or provide details of the exact calculation. E.g., if it requires cut-off rigidity where do you get that number from?

Apologies for the incorrect doi, we have updated the manuscript to provide the correct DOI for this document. To our knowledge this is the known method to obtain the beta coefficient. The cut-off rigidity is taken from the networks themselves or obtained using tools supplied by Hydroinnova at [www.crnslab.org](www.crnslab.org) with the reference describing the tool being:

Desilets, D., 2021, Cutoff rigidity calculations for cosmic ray neutron sensors, Hydroinnova Technical Document 21-02, doi: 10.5281/zenodo.5396587.

Line 245: Please clarify again the difference between the two correction factors fi and fi' for incoming intensity.

Added sentence to clarify the key difference between the two correction factors

Filter: When do you apply the filters, before or after correction of Nraw?

The QA flags are applied both before and after correction depending on the flag. We have updated the manuscript to describe this.

Filter: very often sensors have maintenance periods where the data is not to be trusted. Does crspy support manual definitions of to-be-excluded periods?

Not at this stage, however this is an idea that could be implemented in the future as we endeavour to update crspy with user feedback.

Line 353: this sentence has a circular problem: "Each site is also be given a country code and a site number in the metadata, which is used by crspy to find any required values stored in the metadata"

We have re-worded the sentence to be clearer.

Figure 2: hourly time steps are hard to visualize for long time series given the high fluctuations, and monthly aggregation looks very abstract. I'd recomment to use daily aggregation for all panels. This would allow for better illustration of the differences between the models in panels a–d, and it would no longer require panels e–f.

We thank the reviewer for this suggestion. We have updated the figure so that now it shows daily aggregated data in place of hourly data, which reduces the noisiness of the first 4 charts. We have decided to keep in panels e-f also as we believe this still adds value in demonstrating how mean values differ for longer aggregation of soil moisture.

**Appendix**

important note: require date time to be in UTC

Corrected to state UTC time

explain what unmoderated and moderated means

Added a brief description on this

What if sensors have more than 2 or less than 2 pressure sensors?
Currently we have designed crspy using current networks as a base format. We noticed that many of these networks had up to 2 pressure sensors. If there are less than two crspy will use the one that is available. If there are more than two then this is currently outside the scope of crspy, however this can be adjusted if it was required in the future.

We once again would like to thank all the reviewers and the editor for taking the time to read and provide valuable feedback for this manuscript.

**References**

Cooper, H. M., Bennett, E., Blake, J., Blyth, E., Boorman, D., Cooper, E., Evans, J., Fry, M., Jenkins, A., Morrison, R., Rylett, D., Stanley, S., Szczykulska, M., Trill, E., Antoniou, V., Askquith-Ellis, A., Ball, L., Brooks, M., Clarke, M. A., Cowan, N., Cumming, A., Farrand, P., Hitt, O., Lord, W., Scarlett, P., Swain, O., Thornton, J., Warwick, A., and Winterbourn, B.: COSMOS-UK: national soil moisture and hydrometeorology data for environmental science research, Earth Syst. Sci. Data, 13, 1737–1757, https://doi.org/10.5194/essd-13-1737-2021, 2021.

Desilets, D., 2021, Cutoff rigidity calculations for cosmic ray neutron sensors, Hydroinnova Technical Document 21-02, doi: 10.5281/zenodo.5396587.

Evans, J. G., Ward, H. C., Blake, J. R., Hewitt, E. J., Morrison, R., Fry, M., Ball, L. A., Doughty, L. C., Libre, J. W., Hitt, O. E., Rylett, D., Ellis, R. J., Warwick, A. C., Brooks, M., Parkes, M. A., Wright, G. M. H., Singer, A. C., Boorman, D. B., Jenkins, A., Evans, J. G., Libre, J. W., Jenkins, A., Rylett, D., Singer, A. C., Warwick, A. C., Morrison, R., Ward, H. C., Ellis, R. J., Ball, L. A., Hewitt, E. J., Fry, M., Parkes, M. A., Boorman, D. B., Hitt, O. E., Brooks, M., Wright, G. M. H., and Doughty, L. C.: Soil water content in southern England derived from a cosmic-ray soil moisture observing system - COSMOS-UK, 30, 4987–4999, https://doi.org/10.1002/hyp.10929, 2016.

Hawdon, A., Mcjannet, D., and Wallace, J.: Calibration and correction procedures for cosmic-ray neutron soil moisture probes located across Australia, 1–17, https://doi.org/10.1002/2013WR014333.Received, 2014.

Hands, A. D. P., Baird, F., Ryden, K. A., Dyer, C. S., Lei, F., Evans, J. G., Wallbank, J. R., Szczykulska, M., Rylett, D., Rosolem, R., Fowler, S., Power, D., and Henley, E. M.: Detecting

Ground Level Enhancements Using Soil Moisture Sensor Networks, Space Weather, 19, https://doi.org/10.1029/2021SW002800, 2021.

Zreda, M., Shuttleworth, W. J., Zeng, X., Zweck, C., Desilets, D., Franz, T., and Rosolem, R.: COSMOS: The cosmic-ray soil moisture observing system, 16, 4079–4099, https://doi.org/10.5194/hess-16-4079-2012, 2012.

---

## Author Response (AR2)

Dear Editor,

Firstly, we would like to extend our thanks to yourself and the three anonymous reviewers, whose time and valuable feedback have helped to improve this manuscript.

We have updated the manuscript to include the analysis on the benefits of using ERA5-Land data within the main body of the manuscript and expanded the initial figure. Please also note that the software version has gone from v1.2 to v1.2.1 to reflect some minor changes made within the code whilst we revised the manuscript (e.g. addressing minor bugs or typos). The zenodo dois have been updated to reflect this.

Kind Regards,

Dan Power (on behalf of all coauthors)